# Antibiotic resistance selection and deselection in municipal wastewater from 47 countries

Zhuofeng Yu [1,2], Declan A. Gray[1,2,3], Jerker Fick [4], Noel Waters [1,2], Richard Lindberg[4], Roman Grabic [5], Mats Tysklind [4], Mutshiene Deogratias Ekwanzala [1,2], Hannah-Marie Martiny [6], Carl-Fredrik Flach [1,2], Frank M. Aarestrup [6] & D. G. Joakim Larsson [1,2] ✉

Wastewaters are plausible arenas for antibiotic resistance evolution and transmission, yet selection for resistance by municipal wastewater has rarely been empirically demonstrated. Here, we experimentally investigate the potential of untreated municipal wastewater from 47 countries to select for antibiotic resistance and explore possible drivers. Using a functional selection assay with 340 mixed *Escherichia coli* strains, we find that sterile-filtered samples from 14 countries select significantly for resistance to at least one antibiotic class compared to baseline, while the majority select significantly against resistance. Similar results were generated using natural wastewater communities. Additionally, we report a comprehensive characterization of antibiotics and antibacterial biocides in the wastewaters. None of the 22 analyzed antibiotics could be assigned as key drivers for selection in *E. coli*, whereas e.g. folate pathway antagonists and macrolides often exceed predicted non-selective concentrations for other bacteria by >10-fold. Despite detecting 13 (out of 20 analyzed) organic antibacterial biocides, their potential for co-selection remains unclear. Measured chemical constituents correlate only weakly with observed selection, suggesting complex mixture effects and/or selection by unmeasured compounds. The clear deselection of resistance by most samples indicates that many resistant strains have impaired fitness in wastewaters with limited antibiotic selection pressure.

Antibiotic resistance is a pressing challenge to public health, domestic animal health, and food production systems globally[1–3]. The widespread use and misuse of antibiotics have accelerated the emergence of resistant pathogens, with alarming consequences for the management of bacterial infections[4,5]. Given that no new classes of broad-spectrum antibiotics that are effective against Gram-negative bacteria have reached the market in decades[6,7], there is an increasing need to step up actions to slow down both the emergence of new forms of resistance in pathogens as well as their spread.

[1]Centre for Antibiotic Resistance Research (CARe) at the University of Gothenburg, Gothenburg, Sweden. [2]Department of Infectious Diseases, Institute of Biomedicine, The Sahlgrenska Academy at the University of Gothenburg, Gothenburg, Sweden. [3]The Sahlgrenska University Hospital, Gothenburg, Sweden. [4]Department of Chemistry, Umea University, Umeå, Sweden. [5]Research Institute of Fish Culture and Hydrobiology, South Bohemian Research Center of Aquaculture and Biodiversity of Hydrocenoses, Faculty of Fisheries and Protection of Waters, University of South Bohemia in České Budějovice, Vodňany, Czech Republic. [6]Research Group for Genomic Epidemiology, National Food Institute, Technical University of Denmark, Kgs. Lyngby, Denmark. ✉e-mail: joakim.larsson@fysiologi.gu.se

Selection pressure by antibiotics is a key driver of resistance evolution[8]. Most efforts globally have been and are still targeted at reducing unnecessary selection on the microbiota of humans and domestic animals. However, in recent years, the role of the external environment as a source and arena for resistance evolution has become increasingly recognized[9,10]. If some of the evolutionary steps that lead to mobilization and eventually the emergence of resistance in bacterial pathogens occur in the external environment, actions to limit antibiotic exposure in humans and domestic animals may be insufficient.

Municipal wastewater provides an environment where many pathogens and other human-adapted bacteria meet diverse environmental species in the presence of a mixture of antibiotics and other chemicals. Recent genetic evidence also identifies municipal wastewater as a plausible arena both for the mobilization[11,12] and the horizontal transfer[13] of antibiotic resistance genes (ARGs). Bacterial species identified as recent chromosomal origins of contemporary mobile ARGs typically lack the specific mobilization machinery—namely insertion sequences (IS) and insertion sequence common regions (ISCRs)—that were presumably instrumental in the initial mobilization of these corresponding ARGs[11]. However, known origin species and corresponding IS/ISCRs frequently occur together in high abundances in wastewaters, less so than in human microbiota or soils[11,12]. Furthermore, a recent study pinpointed wastewater as an important environment for horizontal transfer of resistance based on the co-occurrence of taxa carrying the same ARGs[13]. In line with these observations, the plasmidome of bacteria in municipal wastewater systems is highly diverse and rich in ARGs[14].

Whether the concentrations of antibiotics in domestic wastewaters are sufficiently high to create selection pressure is, however, not clear. Studies simply comparing relative ARG abundances in influent and effluent metagenomes do not provide conclusive evidence of selection, as changes in species composition that have nothing to do with antibiotic selection pressures often have a large influence on ARG counts. Most field studies addressing within-species selection at municipal wastewater treatment plants (WWTPs) are limited in different ways, most often by the analysis of a limited number of isolates[15–17]. The most comprehensive study to date comparing phenotypic resistance rates in a bacterial species (*Escherichia coli*) between the influent and the effluent of a WWTP did not suggest any clear selection for antibiotic resistance[15]. However, it should be noted that this was a Swedish site, a country with internationally low antibiotic consumption[18]. Hence, it is certainly possible that there could be selection for resistant strains at other WWTPs, particularly in countries with higher use of antibiotics.

There is an apparent option to assess the potential for resistance selection by comparing measured antibiotic concentrations in wastewaters to concentrations known or predicted to select for antibiotic resistance[19,20]. However, this approach has some limitations. First, it is difficult or currently impossible to accurately analyze all antibiotics in a complex matrix such as wastewater[21]. Second, for most antibiotics, available data on selective potential is still simply a prediction based on the potential of the antibiotics to affect the growth of bacteria[20]. For those few antibiotics for which there is actual data on within-species selection potential, selective concentrations may vary greatly depending on which assay was applied[22,23]. Third, other substances, such as antibacterial biocides, might contribute to antibiotic resistance selection through co-selection, but for these, there is even less information on co-selective potency[24,25]. Fourth, such an approach would not be able to capture interactions (additive, synergistic, or antagonistic) between different antibiotics or between antibiotics and other known or unknown components of wastewater. This calls for an approach that investigates the total selection pressure of the entire mixture of chemicals present in municipal wastewater, as was recently done for hospital wastewater by Kraupner et al.[26].

The Global Sewage Surveillance Project has coordinated the sampling of influents to WWTPs worldwide, with the primary intention of characterizing municipal sewage microbial communities through metagenomic analyses, particularly regarding antibiotic resistance[27,28]. Here, we aimed to investigate the potential of untreated municipal wastewater sampled from 47 countries to select for antibiotic resistance in *E. coli*. For this purpose, wastewater samples from a recent global sampling campaign[29] were sterile-filtered and assessed for selection potential by passaging a synthetic community of 340 environmental *E. coli* strains, followed by culturing on selective agar. For a subset of samples, assays with natural wastewater communities were used as a complement. In parallel, antibiotics and antibacterial biocides were analyzed by online solid phase extraction liquid chromatography tandem mass spectrometry (OSPE-LC-MS/MS) in all samples. Chemical concentrations were compared to data on selection potential as well as relative ARG and biocidal resistance gene (BRG) abundances in the corresponding unfiltered samples, as reported elsewhere[29]. The study demonstrates a clear selection pressure for antibiotic resistance caused by some municipal wastewaters, but for the majority of samples, clear selection against resistance is found. Together, this adds to our understanding of the risks of resistance evolution and transmission in municipal wastewater. Given that only one composite sample per country was analyzed in most cases, we refrain from concluding about country-specific risks.

## Results

### Selection potential for antibiotic resistance in globally sourced municipal wastewater samples

The selection potential of 49 sterile-filtered municipal wastewater samples from 47 countries across the world was assessed using a synthetic community of 340 mixed, *E. coli* strains, each collected from wastewaters with a strategy to generate an assemblage with highly diverse resistance profiles (Waters et al. in preparation; see also Methods). Selection potential was measured as the change in the proportion of bacteria resistant (%resistance) to five different antibiotics (amoxicillin/clavulanic acid, ciprofloxacin, cefotaxime, sulfamethoxazole/trimethoprim, and tobramycin, representing major antibiotic classes) after three passages (72-h) of wastewater exposure in 10% LB-medium, compared to the initial %resistance to the same antibiotic before exposure (0-h, baseline) (Fig. 1). The selection potential data presented in Fig. 2 is the log10-transformed value of the resistance ratio (%resistance relative to the baseline). Detailed data are provided in Supplementary Data 1. Ciprofloxacin (10 μg/L) and wastewater from the Sahlgrenska University Hospital in Gothenburg, Sweden (termed "Hospital WW Sweden"), which previously have been shown to select strongly for multi-resistance[26], were included as positive controls and both showed clear positive selection for resistance to all investigated antibiotics (Fig. 2). Fourteen municipal wastewater samples originating from Algeria, Benin, Denmark, Germany, Lebanon, Luxembourg, Nigeria, Poland, Slovakia, South Africa, Spain, USA Wisconsin, the United Arab Emirates and the United Kingdom, selected significantly for resistance to at least one antibiotic ($p < 0.05$, two-sided Wald test with Benjamini–Hochberg adjustment). The municipal wastewater sample from Nigeria caused significant selection for all five tested antibiotics, whereas the other samples only selected significantly for resistance to one or two antibiotics ($p < 0.05$, two-sided Wald test with Benjamini-Hochberg adjustment).

An increase in %resistance in relation to the initial %resistance is sometimes referred to as positive selection, whereas an increase in relation to parallel control in time has been referred to as "increased persistence"[30]. When using saline (72-h) as an alternative reference control, nine additional samples (beside the 14 samples shown in Fig. 2) from Austria, Bosnia and Herzegovina, Canada, Finland, Hungary, Malawi, Serbia, Slovenia, and USA Washington significantly selected for resistance ("increased persistence") to at least one

 

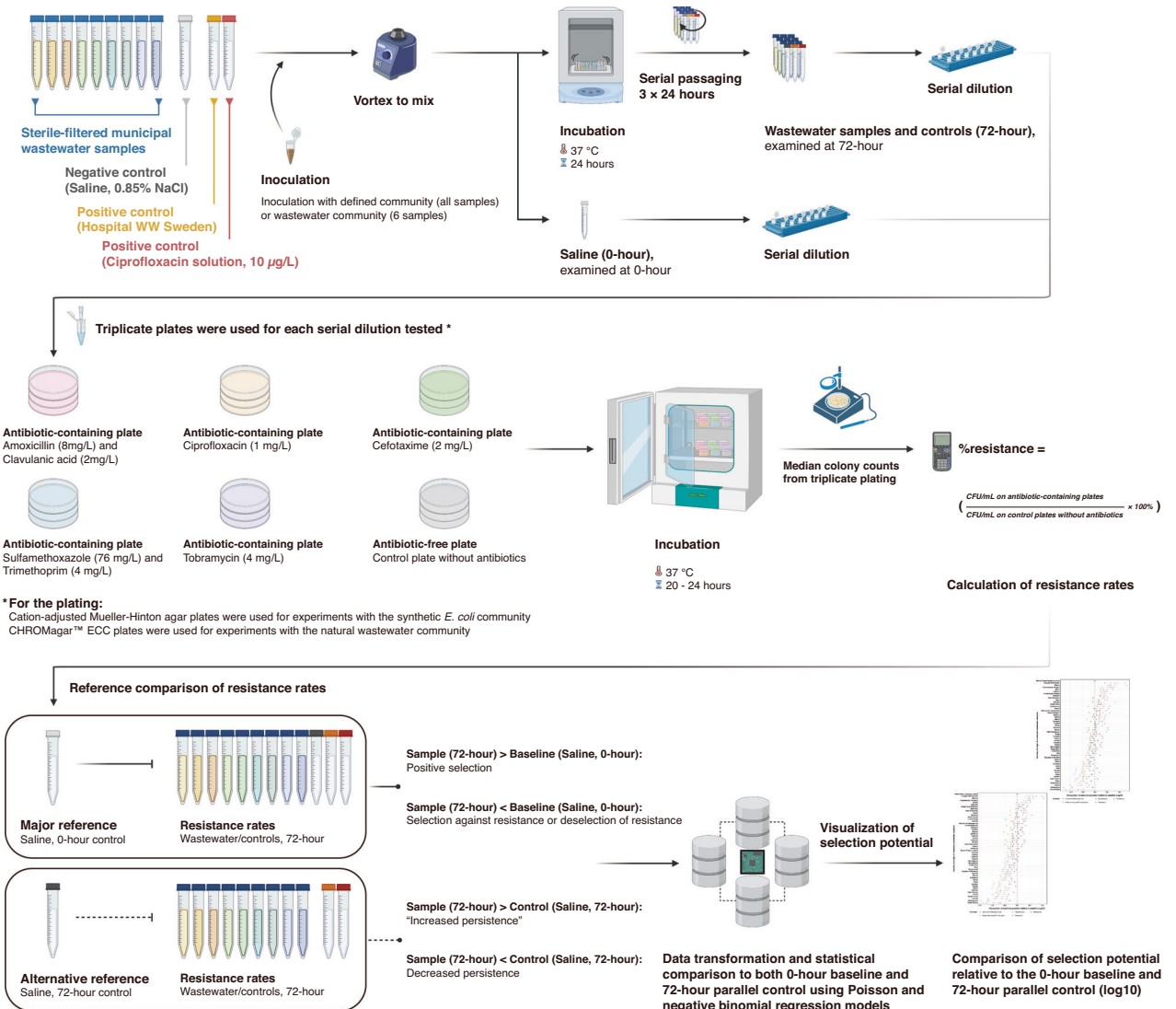

**Fig. 1 | Workflow to quantify selection potential for antibiotic resistance in *E. coli* in globally sourced municipal wastewater samples.** Sterile-filtered untreated municipal wastewater samples ($n = 49$, nine samples shown for illustration purposes), positive controls (wastewater from the Sahlgrenska University Hospital in Gothenburg, Sweden, termed "Hospital WW Sweden"; ciprofloxacin, 10 μg/L#) and a negative control (saline, 0.85% NaCl) were mixed with LB medium (10%) to a final volume of 3 mL. Each sample was then inoculated with a synthetic mixed community of diverse *E. coli* (340 strains). For additional validation, six wastewaters ( + controls) were inoculated with a natural wastewater community. The cultures were passaged at 24-h and 48-h intervals by transferring 3 μL of the grown cultures to fresh tubes containing 3 mL of the wastewater samples/controls with 10% LB medium. Nine to fifteen wastewater samples were tested in parallel in each experiment; hence, it took four experiments to cover all wastewaters once. Then, the entire workflow (and analyses) was repeated three additional times. Baseline samples (0-h) as well as serially passaged cultures (72-h) were plated onto agar with and without antibiotics. Colony-forming units (CFUs) of *E. coli* were enumerated on cation-adjusted Mueller-Hinton or CHROMagar™ ECC plates in triplicate for the synthetic *E. coli* community and the natural wastewater community, respectively. The resistance rate (%resistance) for each antibiotic was calculated based on the median colony counts on antibiotic-containing *versus* antibiotic-free plates. Selection may be evaluated in relation to different controls[9]. Changes compared to the 0-hour baseline control are here termed selection/deselection, while changes compared to the 72-hour parallel control are termed increased or decreased persistence, respectively. Changes in %resistance were evaluated statistically and visualized as fold-change (log10) in relation to the 0-hour baseline (Fig. 2) or the 72-h parallel control (Supplementary Fig. 1). Figure 1 was created in BioRender. Larsson, J. (2025) https://BioRender.com/48u2259 # This represents the final ciprofloxacin concentration (10 μg/L) after mixing with the LB medium in the testing system.

antibiotic. In samples from Lebanon and the United Arab Emirates, resistance to all antibiotics were selected for (Supplementary Fig. 1, Supplementary Data 2).

Despite several examples of significant positive selection, most of the wastewater samples showed no antibiotic resistance selection. On the contrary, in the majority of samples (40 out of 49), at least one type of resistance was significantly selected against, i.e., deselected ($p < 0.05$, two-sided Wald test with Benjamini-Hochberg adjustment). For samples from eight countries (Ghana, Brazil, Sweden, China, South Korea, Iceland, Madagascar, and Saudi Arabia), significant deselection was observed for resistance to all antibiotics (Fig. 2). Also, exposure to saline for 72 h significantly deselected resistant strains, but considerably less so than exposure to many wastewater samples.

The observed selection and deselection patterns followed similar overall trends across all antibiotic resistance phenotypes but with clear variations between samples. The strongest and most common deselection was observed for resistance to sulfamethoxazole/trimethoprim and ciprofloxacin, whereas most cases of positive selection were observed for tobramycin and amoxicillin/clavulanic acid resistances.

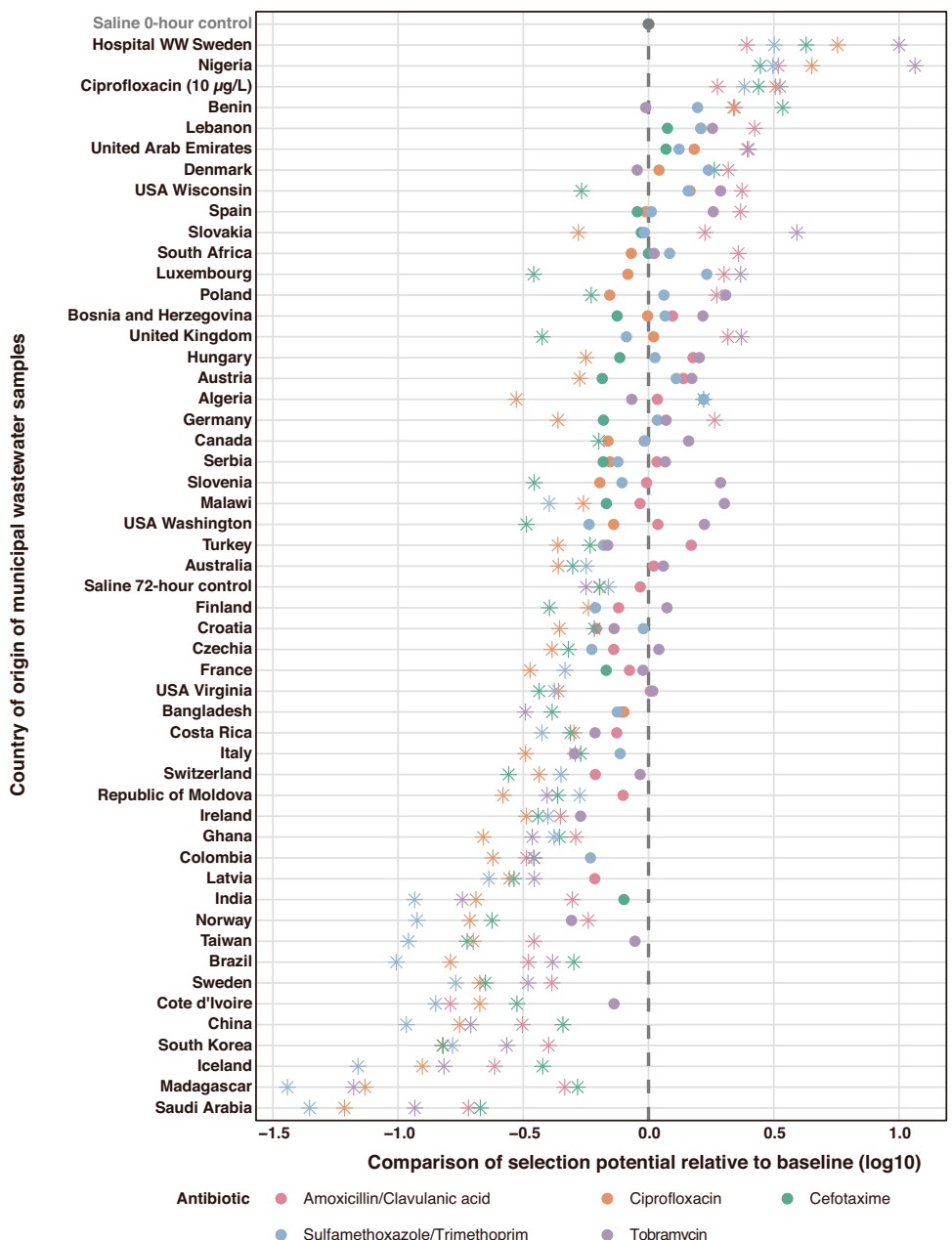

**Fig. 2 | Selection potential of globally sourced municipal wastewater for different antibiotic resistances.** The selection potential represents the %resistance in an *E. coli* community exposed to globally sourced municipal wastewater samples [and ciprofloxacin (10 μg/L) and wastewater from a Swedish hospital as positive controls, and saline as negative control] after three passages (72-h) relative to the initial %resistance (0-h, baseline; dashed gray line). The selection potential of individual resistances relative to the baseline is log10-transformed to allow resistance to different antibiotics to be viewed at a comparable scale. Data points to the right of the baseline indicate positive selection, whereas those to the left indicate deselection. Countries and controls are ordered based on the mean selection potential for all antibiotics. Three samples were collected from locations in the United States of America (states indicated). Selection potentials that are not significantly different from baseline (0-h) are depicted as solid circles, while those with statistically significant differences ($p < 0.05$, two-sided Wald test with Benjamini-Hochberg adjustment; see Supplementary Data 1 for the detailed $p$ values) are marked with asterisks. See Supplementary Fig. 1 for an alternative comparison to the 72-h saline control.

## Validation of selection findings with natural wastewater communities

To validate our findings from the synthetic *E. coli* community and to assess potential experimental artifacts, we conducted additional selection assays using natural wastewater microbial communities for a subset of samples. We selected municipal wastewater samples from six countries (Benin, Lebanon, Madagascar, Nigeria, South Africa, and the United Arab Emirates) based on their varying selection potentials in

the first run of the selection experiment and detected antibiotic concentrations (see below), along with the same control exposures as before. The proportional changes of resistant bacteria at 72-hour were compared between the synthetic *E. coli* and natural wastewater communities (Fig. 3). Despite very different communities, overall results were congruent between the two assays, with samples that showed strong positive selection in the synthetic *E. coli* community also doing so in the natural community, while deselection results also matched

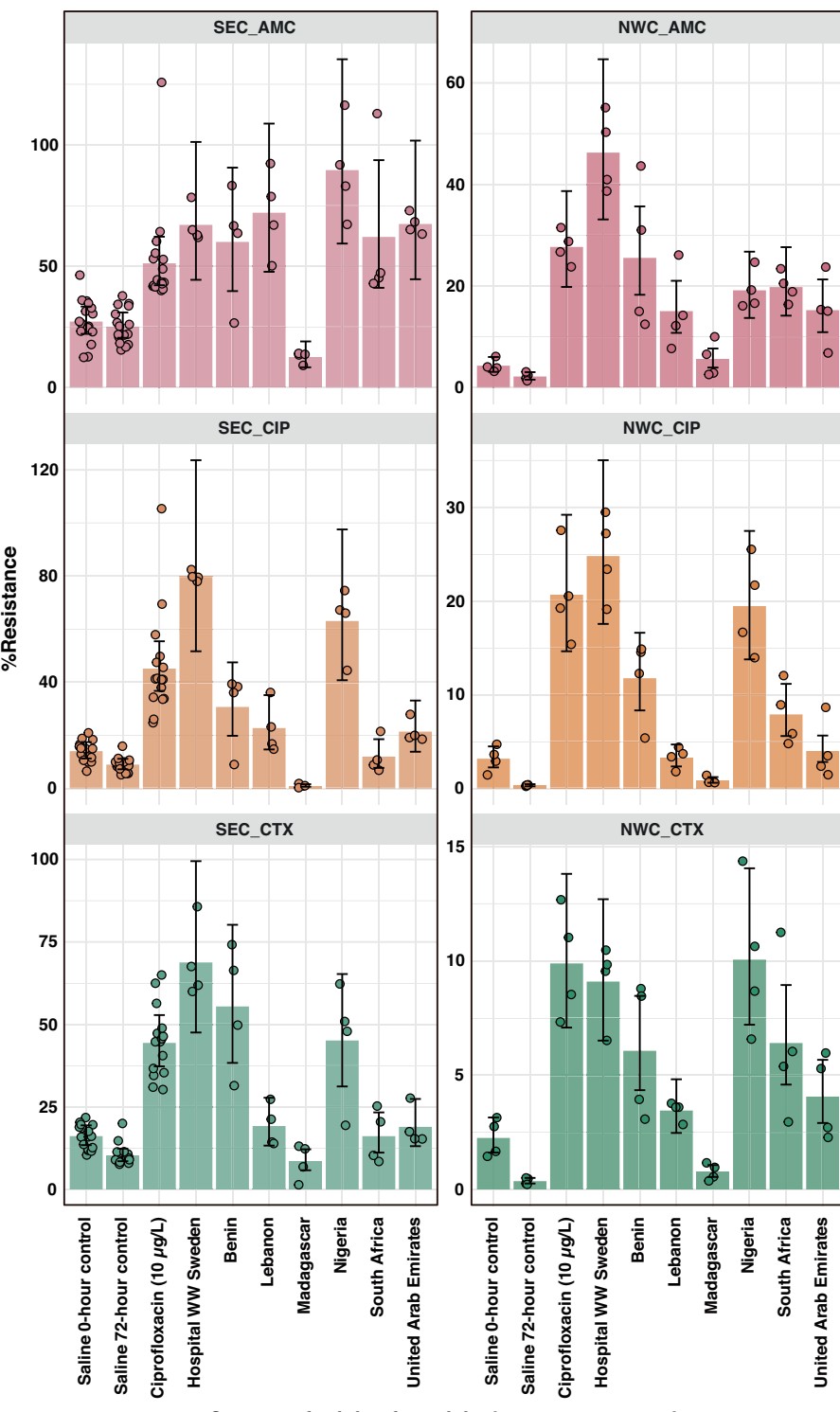

**Fig. 3 | Comparison of selection potential of different wastewaters in *E. coli* assessed using either single- or multi-species communities.** Bars represent the percentage of *E. coli* resistant to three antibiotics (top to bottom) in a synthetic *E. coli* community (left) and a natural wastewater community (right) after three passages (72 h) in the presence of sterile-filtered wastewater from Benin, Lebanon, Madagascar, Nigeria, South Africa, and the United Arab Emirates, along with control conditions [saline, ciprofloxacin (10 µg/L), and wastewater from a Swedish hospital]. Data are presented as estimates ± confidence intervals of %resistance for each sample from the regression models (see Methods, and also Supplementary Data 1 and Supplementary Data 3). Specifically, the error bars indicate the lower 95% and upper 95% confidence intervals for %resistance. Four data points are shown for each sample/control, representing the direct measurements of %resistance from the four independent runs of selection assays[#]. Abbreviations: SEC, test with the synthetic *E. coli* community; NWC, test with the natural wastewater community; AMC, %resistance to amoxicillin/clavulanic acid; CIP, %resistance to ciprofloxacin; CTX, %resistance to cefotaxime. [#] In the selection assay employing SEC and NWC, all municipal wastewater samples and "Hospital WW Sweden" were analyzed across four independent experiment runs, yielding four replicates per sample per community. Saline and ciprofloxacin (10 µg/L) controls were included in each of the four discrete experiments within every run when using SEC, generating more replicates ($n = 16$) compared to wastewater samples and other control conditions.

well. This validation indicates that our observations were not artifacts of the synthetic community approach and suggests that the selection observed would likely apply to diverse microbial populations in real wastewater communities as well. In Supplementary Fig. 2, the selection observed in the natural wastewater community is illustrated similarly to that done in Fig. 2 for the synthetic community, with detailed selection data presented in Supplementary Data 3.

## Effects of sample processing and storage on selection potential

To assess whether the prolonged storage and shipping conditions of the wastewater samples might have affected their selection potential, we conducted a separate experiment using fresh wastewater from Sahlgrenska University Hospital, Gothenburg, Sweden. This wastewater was processed similarly to the Global Sewage Surveillance Project protocol[27], albeit with shorter storage times, and tested for selection potential at different steps during processing (Supplementary Fig. 3). Storage and sample processing notably affected the selection potential, with a general trend toward reduced selection over time (Supplementary Fig. 4). Selection against all tested antibiotics was significantly higher ($p < 0.05$, one-way ANOVA with Tukey's test adjustment) in immediately processed fresh hospital wastewater (Test 1, representing the ideal situation) compared to the same wastewater subjected to conditions simulating those relevant for the globally sourced municipal wastewater samples (Test 5). Notably, this was a hospital wastewater sample with an initially strong selection potential, and it maintained a significant selection potential even after prolonged storage (Test 5 compared to the saline 0-h control, $p < 0.05$, one-way ANOVA with Tukey's test adjustment). In parallel, we monitored antibiotic concentrations, suggesting gradual but moderate reductions in the concentrations of most antibiotics (Supplementary Fig. 5, Supplementary Data 4). Together, these suggest that while our measurements of selection potential in the wastewater samples might be conservative estimates due to potential antibiotic degradation during storage, any findings of positive selection likely represent real selection potential of such wastewaters. Accordingly, it is possible that deselection could have been less pronounced in some cases if fresh samples had been available and tested.

## Antibiotic concentrations in relation to selective concentrations

The concentrations of 22 antibiotics were measured in all samples by OSPE-LC-MS/MS: 21 antibiotics were detected, while ampicillin was below the limit of quantification (LOQ) in all samples, consistent with previous research indicating its rare occurrence in municipal WWTP influent[21]. As indicated by the dashed reference lines in Fig. 4, measured concentrations were generally far below the lowest one percentile of reported minimum inhibitory concentrations (1% lowest MICs) for *E. coli* reported in the EUCAST database (v. 2024-05-13, Supplementary Data 5). The 1% lowest reported MIC for any bacteria (including an additional safety factor) is often used as a reference for risks of resistance selection[20,23,31,32], hence the analogy of using MIC data only for *E. coli* here. Two exceptions were observed with possibly more potent concentrations: the sample from South Africa, where ciprofloxacin concentrations approaching the 1% lowest MIC for *E. coli*, as indicated by a dashed line (positive selection observed for amoxicillin/clavulanic acid only); and the sample from South Korea, where norfloxacin concentrations exceeded this threshold (strong deselection for resistance to all tested antibiotic classes). Sulfamethoxazole, an inexpensive and widely available antibiotic, showed generally higher concentrations in African samples than in other regions, whereas meropenem, an expensive intravenous antibiotic, was not detected in any African samples (Fig. 4). These findings align with a recent review article highlighting sulfonamide antibiotics as particularly common contaminants in African waters, in comparison with other continents[33]. When compared to predicted no-effect concentrations (PNEC) for general species (based on 1% lowest MICs for any species) as reported by Bengtsson-

Palme and Larsson[20], several samples contained antibiotic concentrations exceeded PNECs for resistance selection. Ratios of measured antibiotic concentration to the PNEC for general species are represented by a color gradient in Fig. 4. Raw data for antibiotic concentrations, including the indicated 1% lowest MIC and PNEC values for general resistance selection, are provided in Supplementary Data 6.

## Global distribution of antibacterial biocides in municipal wastewater samples

We have quantified a broad set of antibacterial biocides in municipal wastewaters across many countries and continents. The analysis revealed the presence of diverse biocides spanning different chemical classes, including acids, aldehydes, biguanides, phenolic compounds, polyamines, and quaternary ammonium compounds (QACs), as illustrated in Fig. 5. Among these, acids were detected at notably elevated concentrations relative to other biocide categories. However, most biocides were detected at relatively low concentrations, with approximately 50% of the analyzed biocides falling below LOQs (Supplementary Data 7). As co-selective concentrations for antibacterial biocides are largely lacking, a systematic comparison with PNECs, as was done for the antibiotics above, could not be conducted.

## Regression analysis relating chemical concentrations to selection potential and resistance genes

To more exploratively identify potential drivers of selection, we investigated the relationship between antibiotic concentrations and biocide concentrations and the observed selection rates (i.e., %resistance) after exposure. Additionally, we examined how well the variance in the relative ARG and BRG abundances using sequencing data from our previous study[29] was associated with the chemical concentrations measured in the same samples here, as a strong selection pressure might influence resistance gene counts (although apparently also heavily influenced by variable resistance in incoming bacteria)[34]. In this analysis, we present Benjamini-Hochberg corrected $p$ values, denoted as $q$ values (Supplementary Figs. 6–16).

Folate pathway antagonist and macrolide concentrations maintained significant positive associations with selection rates for multiple antibiotics ($q = 0.02$–0.03), but with moderate $R^2$ values (0.15–0.20) (Supplementary Fig. 6). Both trimethoprim and sulfamethoxazole were significantly associated with resistance selection against at least one antibiotic class ($R^2 = 0.12$–0.17, $q = 0.02$–0.04), whereas azithromycin was the solely responsible macrolide compound for the observed associations with selection rates ($R^2 = 0.12$–0.20, $q = 0.02$–0.04) (Supplementary Fig. 7). For the biocides, only the acid group was significantly associated with observed selection rates, but was associated with resistance to four antibiotics: amoxicillin/clavulanic acid, ciprofloxacin, sulfamethoxazole/trimethoprim and tobramycin (Supplementary Fig. 8). The observed association with acids was driven by salicylic acid but not by 1H-benzotriazole (Supplementary Fig. 9).

Regression analysis between antibiotic concentrations and relative ARG abundances demonstrated fewer positive associations (Supplementary Fig. 10), with folate pathway antagonists significantly associated with tetracycline resistance genes ($R^2 = 0.27$, $q = 0.006$) and macrolide concentrations notably associated with beta-lactam resistance genes ($R^2 = 0.26$, $q = 0.006$). Further analysis revealed that within the folate pathway antagonists, sulfamethoxazole alone accounted for the observed association ($R^2 = 0.27$, $q < 0.001$, Supplementary Fig. 11). Similarly, among the macrolides, clarithromycin was the sole contributor to the association ($R^2 = 0.19$, $q = 0.005$, Supplementary Fig. 12). In contrast, our analysis demonstrated that antibiotic concentrations were not significantly associated with relative BRG abundances (Supplementary Fig. 13). Additionally, we found no significant associations between the concentrations of any biocide groups and either ARGs or BRGs (Supplementary Figs. 14, 15).

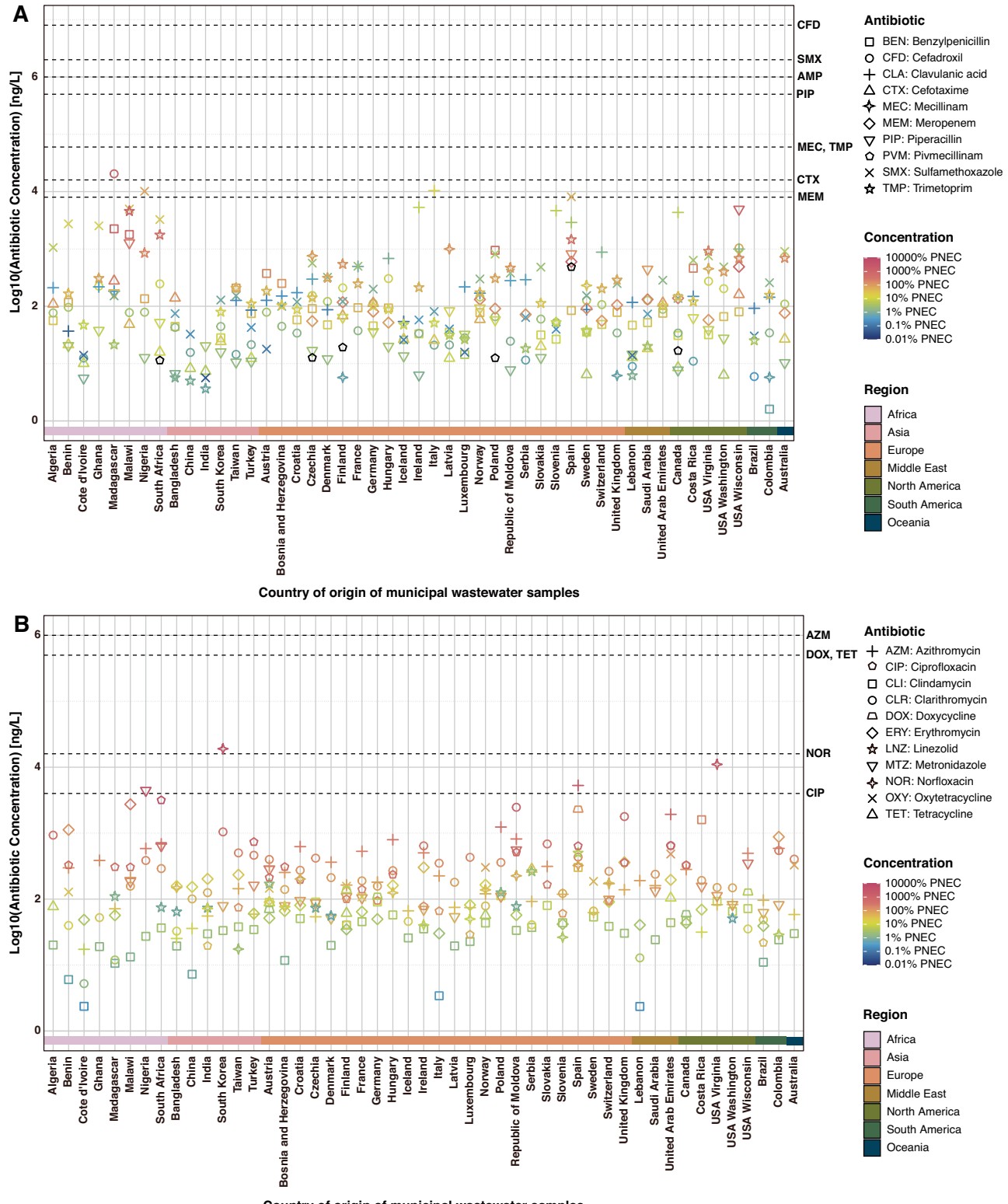

**Fig. 4 | Antibiotic concentrations in globally sourced municipal wastewater samples. A** Antibiotic concentrations (ng/L) of beta-lactams (ampicillin, benzyl-penicillin, cefadroxil, cefotaxime, clavulanic acid (inhibitor), mecillinam, meropenem, piperacillin, pivmecillinam) and folate pathway antagonists (sulfamethoxazole, trimetoprim). **B** Antibiotic concentrations (ng/L) of lincosamides (clindamycin), macrolides (azithromycin, clarithromycin, erythromycin), nitroimidazoles (metronidazole), phenicols (linezolid), quinolones (ciprofloxacin, norfloxacin) and tetracyclines (doxycycline, oxytetracycline, tetracycline).

Different antibiotics are represented by distinct symbols. A color gradient is applied to these symbols based on the ratio of antibiotic concentration to the non-species specific predicted no-effect concentration (PNEC) for resistance selection as reported by Bengtsson-Palme and Larsson[20]. Pivmecillinam is depicted in black because the corresponding PNEC data are unavailable. The dashed lines indicate the 1% lowest MICs for *E. coli* in the EUCAST database, indicative of selection potential in *E. coli* specifically.

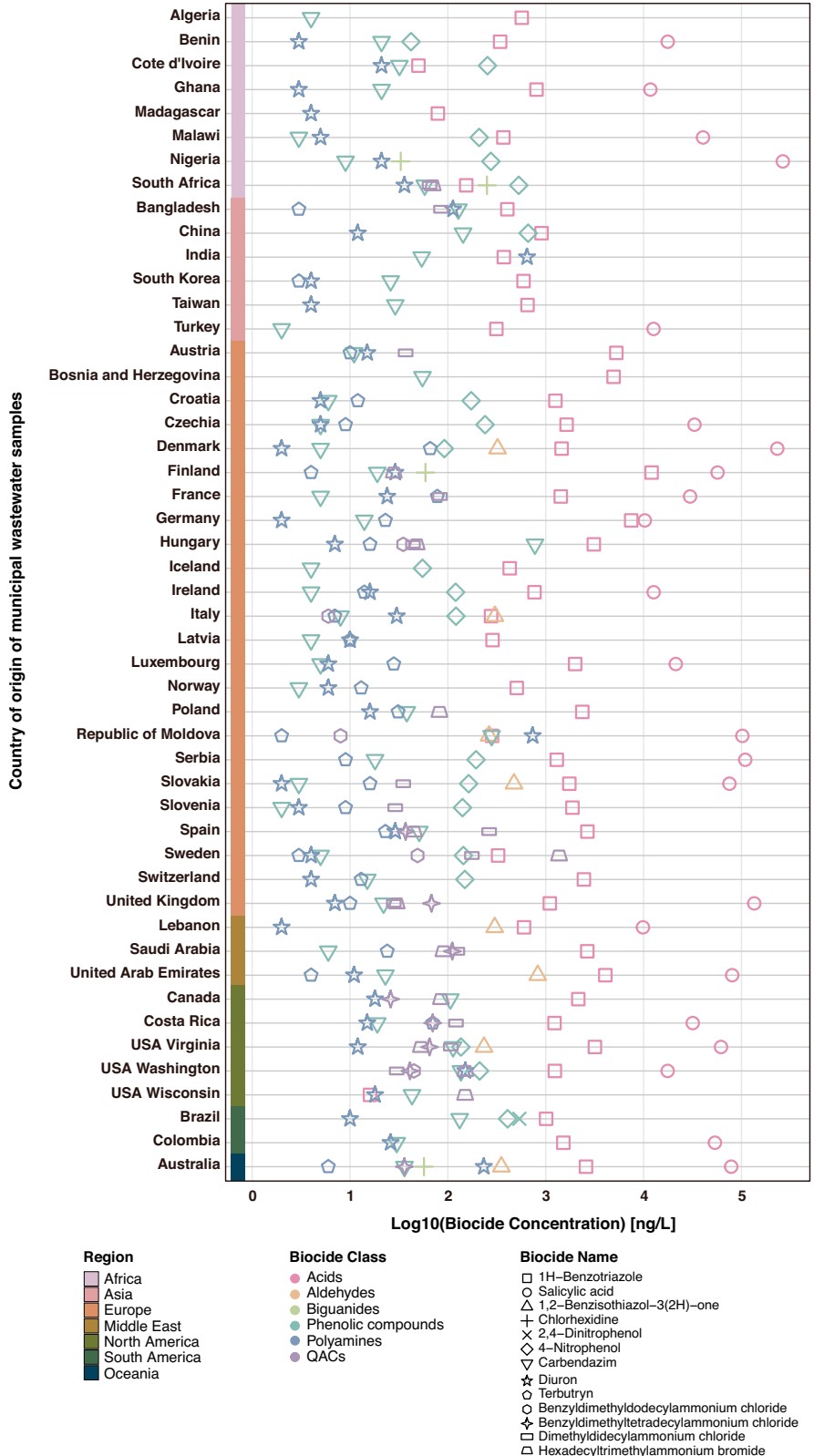

**Fig. 5 | Antibacterial biocide concentrations in globally sourced municipal wastewater samples.** Concentrations (ng/L) of six antibacterial biocide classes (acids, aldehydes, biguanides, phenolic compounds, polyamines, QACs) are presented. The color scheme denotes distinct biocide classes.

As dilution in the sewer pipes is likely to matter, we also investigated the relationship between reported national municipal water consumption levels per capita against observed selection rates, measured concentrations of antibiotics and biocides, as well as relative abundances of ARGs and BRGs (Supplementary Fig. 16). No significant associations were identified between water consumption and the observed selection, biocide concentrations, or BRGs. There was, however, a significant association between water consumption and three classes of ARGs (folate pathway antagonist, nitroimidazole, and phenicol) with *q* values ranging from <0.001 to 0.03, as well as

against concentrations of the lincosamide antibiotic, clindamycin ($q = 0.03$).

## Discussion

In this study, we investigated the potential of untreated municipal wastewater from 47 countries to select for antibiotic resistance. We report experimental evidence both for clear positive selection of resistance as well as for selection against resistance to multiple antibiotics. Deselection for antibiotic-resistant *E. coli* was more common than positive selection, possibly linked to the costs of carrying resistance in the absence of strong antibiotic selection pressures, and/or the benefits of specific wastewater adaptations. Some antibiotics were occasionally detected at concentrations above those predicted to be safe for resistance selection in bacteria in general, but far below those known to inhibit growth in *E. coli*. Furthermore, a comprehensive characterization of antibacterial biocides in the municipal wastewaters did not indicate any clear drivers. This suggests either selection by compounds that were not analyzed and/or a result of unknown mixture effects. Despite the unprecedented international coverage and confirmation of results using an independent assay, only one composite sample per country was investigated in most cases. Hence, we refrain from drawing any far-reaching conclusions on selection in specific countries or regions.

The comprehensive characterization of antibiotics and antibacterial biocides provides an important context for interpreting selection results. While the measured concentrations of individual antibiotics were generally far below the 1% lowest MIC reported for *E. coli*, many samples had antibiotic concentrations exceeding general PNECs[20], potentially indicating a higher risk of resistance selection in other bacterial species. For example, ciprofloxacin concentrations ranged from 20 to 3165 ng/L, whereas several studies have reported PNECs or minimal selective concentrations (MSCs) between 64 and 100 ng/L[20,23,35,36]. Seventeen samples had levels that were similar to or exceeded those concentrations. However, only the municipal wastewater sample from Benin (328 ng/L) selected significantly for ciprofloxacin resistance. Notably, the sample with the highest ciprofloxacin concentration (3165 ng/L; South Africa) selected for resistance to amoxicillin/clavulanic acid, but not to ciprofloxacin. Significant selection against ciprofloxacin resistance was observed in many wastewater samples, despite concentrations up to 737 ng/L (Austria, Croatia, Finland, France, Hungary, Madagascar, Malawi, Republic of Moldova, Slovakia, Turkey). Hence, it is plausible that bacterial community dynamics and net resistance selection by complex wastewaters are governed by multiple drivers beyond antibiotic concentrations alone. Azithromycin, sulfamethoxazole, and trimethoprim concentrations showed weak but significant associations with observed selection (Supplementary Fig. 7), while measured concentrations were consistently lower than reported selective concentrations in *E. coli*, often by 1–2 orders of magnitude or more[20,37,38]. This suggests that these specific antibiotics were unlikely to be the primary drivers of the observed selection in our experimental system.

Antibacterial biocides of different chemical classes were also detected. Concentrations of the phenolic compound carbendazim were within the ranges reported in wastewaters from eight Thai WWTPs[39] and ten Chinese WWTPs[40]. The detected QACs, specifically benzyldimethyldodecylammonium chloride and benzyldimethyltetradecylammonium chloride, were approximately 4–5000 times lower than concentrations in river water reported by Chukwu et al.[41] and in WWTP influent by Östman et al.[42]. We cannot exclude that the significant discrepancy may be attributed to the high sorption affinity of QACs for particulate matter, potentially resulting in analyte losses during sample processing[43]. QACs can persist through WWTPs with concentrations ranging from tens of $\mu$g/L to mg/L levels[43], whereas the highest QAC concentration (dimethyldidecylammonium chloride) detected in our municipal wastewater samples was 264 ng/L. Overall,

the absence of established co-selective concentrations of almost all organic biocides limits their interpretability. However, as all measured concentrations were well below those known to inhibit the growth of *E. coli*[44,45], often by several orders of magnitude, co-selection appears unlikely. Salicylic acid was detected at the highest concentrations among the measured biocides, reaching levels up to 265 $\mu$g/L. A significant association was observed between salicylic acid concentration and selective potential for most of the antibiotics tested (Supplementary Fig. 9). However, the lowest MIC for salicylic acid reported for *E. coli* that we are aware of is 310 mg/L[46], which is more than 1000 times higher than the highest detected levels. This substantial difference suggests that salicylic acid is also an unlikely driver of co-selection.

Given the above, the observed positive selection in *E. coli* by some municipal wastewater samples suggests a mixture effect from two or more antimicrobials, a mixture effect between (an) antimicrobial(s) and (an) other factor(s), or simply through drivers that were not analyzed here. It has been shown that an antibiotic mixture, with each component present at a concentration below those significantly affecting growth, may cause large growth inhibition[47]. More investigations into various interactions affecting selection and co-selection potential in wastewaters are therefore warranted.

Our observation that municipal wastewater in many cases selects strongly and broadly against rather than for resistance is an intriguing finding that has not been reported earlier, as far as we are aware of. Such deselection could be attributed to metabolic or other costs associated with maintaining resistance mechanisms in the absence of a strong antibiotic selective pressure[48,49]. In wastewaters with limited antibiotic selection pressure, resistant bacteria may be disfavored when the benefits of resistance no longer outweigh their associated costs. While *E. coli* is primarily an intestinal species, it is becoming increasingly recognized that some *E. coli* thrive in wastewaters and have accordingly adapted genetically to the wastewater environment[50,51]. All the strains in the synthetic community were isolated from wastewater, and the natural community tested was also derived from a municipal WWTP. Consequently, particularly in wastewater environments with low or minimal antibiotic selection pressure, it is possible that wastewater-adapted strains within these communities were preferentially selected over potentially more antibiotic-resistant strains of direct fecal origin. Reciprocally, in wastewater environments with high or elevated antibiotic selection pressure, antibiotic resistance is likely to become the predominant fitness determinant, possibly conferring a selective advantage to clinically significant resistant strains. However, this hypothesis requires further empirical investigation.

The balance between selection and deselection of antibiotic resistance in wastewaters may represent an important ecological dynamic influencing the ability of resistance strains to spread further to other environments, potentially also down the road to the human microbiota. Such a concern is particularly relevant in regions where inadequate sanitation infrastructure plays an important role in the transmission of bacterial diseases[52], and where wastewaters may contribute to the persistence and spread of clinically significant resistant bacterial strains. Future work should explore the mechanisms underlying this deselection phenomenon in various environments, as it may provide insights into ecological interventions that could counter resistance spread.

The high congruence between the selection patterns observed in our synthetic *E. coli* community and the natural wastewater community supports our choice of test system. This consistent pattern indicates that the selection and deselection observed are not artifacts of a particular test system. Whether similar trends in both positive selection, deselection, and absence of net resistance selection translate to real-world field scenarios remains to be investigated. Comparisons of resistance gene abundances in influents with effluents provide very limited evidence of selection/deselection, as taxonomic changes (that

are always large between influents and effluents at WWTPs) are expected to greatly affect ARG abundances[53]. Such between-species selection occurs and has always occurred in almost every context. To assess risks for antibiotic resistance evolution, it is therefore critical to apply tests that can identify selection within species[9,23,37]. Almost all available comparisons of phenotypic resistance in specific bacterial species between influents and effluents from municipal WWTPs are strongly limited, primarily by the number of isolates studied[15–17], and we are only aware of two comprehensive studies. A Canadian study collected 765 *E. coli* isolates from four WWTPs near Montreal, Québec, with 92–100 isolates obtained from both the influent and the effluent of each facility[16]. Frigon et al. randomly selected 719 of these isolates for analysis and showed that the treatment processes notably removed *E. coli* strains that contained multiple virulence factors[16]. Additionally, they observed a significant reduction in *E. coli* carrying the quinolone resistance gene (*qnrS*) as wastewater progressed from influent to effluent[16]. The largest study to date was based on over 4000 *E. coli* isolates collected between September 2015 and March 2017 at the municipal WWTP in Gothenburg, Sweden[15]. No significant change in resistance was found between *E. coli* collected from influents and effluents[15]. In comparison, the Swedish sample analyzed in the present study from the same WWTP (but several years later, in April 2021) showed significant deselection potential. It is quite plausible that the number of generations covered in the current test system (three passages, equaling approximately 30 generations) is considerably higher than that of *E. coli* during the passage through a typical WWTP. Hence, the net effect of either selection or deselection could be smaller in real-life, non-experimental settings. However, there are certainly other species, particularly those thriving in sewage sludge and biofilms, that could be exposed to wastewater for many more generations than *E. coli*, before being released into waterways[54]. In addition, it is also possible that resistant strains could be selected for in species that are more sensitive than *E. coli* to some of the antibiotics present in wastewater. Therefore, more studies on different bacterial species, as well as field studies reflecting the proportion of resistant bacteria in influents and effluents, would be valuable. Depending on how the protocol for antimicrobial resistance surveillance under the recently adopted revision of the Urban Wastewater Treatment Directive will be outlined[55], relevant field data for understanding both selection and deselection may be generated for European WWTPs in the coming years.

The strength of selection/deselection could have been influenced by sample processing and storage conditions. The observed reduction in selection potential and antibiotic concentrations with prolonged storage of hospital wastewater is consistent with previous studies documenting the degradation of some antibiotics during sample storage[56,57]. This suggests that our measurements likely represent conservative (low) estimates of the original actual antibiotic concentrations and the potential to select for resistance.

The choice of control can also influence conclusions regarding selection/deselection. The selection reported here (Fig. 2) represents selection/deselection when comparing %resistance after exposure to the original %resistance at the beginning of the experiments[58], i.e., % resistance exposed to wastewater for 72-hour *versus* at 0-hour. An alternative control would be to compare %resistance after exposure with a parallel control group, i.e., %resistance of wastewater at 72-hour *versus* saline at 72-hour (Supplementary Fig. 1). As the %resistance declined somewhat over time in the community exposed to saline, applying the 72-hour saline control slightly increased the number of samples significantly selecting for resistance, while moderately decreasing the number of samples showing significant deselection. Others have described selection compared to a parallel control in time as "increased persistence" rather than positive selection or enrichment of resistance[30]. Importantly, regardless of which of these controls are applied, the principal findings and interpretations of our study remain unchanged. The lowest antibiotic concentration that maintains resistance ratios to a parallel control group (regardless of changes to the original ratio) has been referred to as the lowest persistence concentration, which has been argued to lead to increasing risks for transmission[30,59]. The concept of using different controls and its implications have been further discussed elsewhere[9,23,37].

There was no clear relationship between municipal water consumption and the observed selection potential, but there was a significant inverse association ($R^2 = 0.17$–$0.33$) between water consumption and several classes of ARGs (Supplementary Fig. 16). Notably, the latter is not evidence of selection in wastewaters, as ARG abundance is likely to reflect the regional resistance situation, at least to some extent[60].

While this study provides clear evidence of selection potential in municipal wastewater, an important remaining knowledge gap includes understanding the drivers. Therefore, parallel studies of selection potential with analyses of additional antibiotics and other potential drivers are needed, as well as more research on interaction effects between such compounds and with other components of wastewater. Second, while we have demonstrated selection in *E. coli* as both a highly relevant pathogen and an environmental bacterium, selection dynamics are likely to differ for other bacterial species. Third, more replication of wastewater analyses within countries would help direct where there are particular concerns for resistance selection in municipal WWTPs. Finally, sources of selective agents cannot with certainty be allocated to emissions from households, as the investigated municipal wastewaters in most if not all cases also receive contributions from other sources, such as hospitals, industries, slaughterhouses and stormwater.

In conclusion, this study represents a comprehensive assessment of selection pressures in municipal wastewaters and provides evidence that untreated municipal wastewater can, in certain contexts, exert a clear and broad selection pressure for antibiotic resistance. Our finding of a clear deselection of resistance for most samples also adds to our understanding of risks for evolution and transmission in municipal wastewater, suggesting that resistant bacteria may be at a competitive disadvantage in many wastewater environments with limited selection pressure.

## Methods

### Global municipal wastewater sample collection and pre-treatment

A total of 49 municipal wastewater samples were collected from 49 cities across 47 countries, representing six major geographical regions (Supplementary Fig. 17). These samples were collected during a campaign conducted as part of the Global Sewage Surveillance Project[29], Supplementary Data 8. A detailed protocol regarding sample collection and processing was documented in a previous study[27]. Briefly, raw wastewater samples were initially consolidated and processed at the National Food Institute, Technical University of Denmark, and the supernatants of refrozen samples were transferred to the University of Gothenburg, Sweden (Supplementary Fig. 18). These samples were thawed in a cold-water bath and transferred on ice before sterile filtration using 0.22 $\mu$m pore size S-Pak filters (Millipore, Bedford, USA). For each sample, aliquots of 200 mL of the filtrate were transferred into two sterile HDPE bottles (Corning, New York, USA) and stored at $-20\,°C$. A 24-h composite wastewater effluent sample from the central complex of the Sahlgrenska University Hospital (Gothenburg, Sweden) was collected on April 28, 2022. This separate sample, i.e., "Hospital WW Sweden", was used as a positive control in the selection experiment (as evidenced by previous documentation[26]) and processed using the same sterile filtration method as the globally sourced municipal wastewater samples. To ensure no bacterial contamination, subsamples of all sterile-filtered samples were incubated at $37\,°C$ for 24 h in 10% LB medium. Subsequently, 100 $\mu$L from each culture was

plated on LB agar and incubated for an additional 24 h at 37 °C, with no bacterial growth detected on any plate.

## Selection potential for antibiotic-resistant *E. coli* in complex communities

To assess the selection potential of the globally sourced municipal wastewater samples, we used a wastewater-derived synthetic *E. coli* community as in Kraupner et al.[26], but more than twice as large. The community used in this study was based on 340 wastewater isolates, selected to represent diverse resistance profiles (Waters et al. in preparation). 320 of these were isolated from a series of 24-h flow-proportional influent samples from the Ryaverket WWTP in Gothenburg, Sweden, using CHROMagar™ ECC plates with no antibiotics or with 23 individual antibiotics. Replica-plating technology was used to isolate strains with resistance to specific combinations of antibiotics while maintaining sensitivity to others. The collection was supplemented with 20 carbapenem-resistant *E. coli* strains collected over several years from both municipal and hospital wastewaters. A pool was prepared by combining 100 μL of each isolate's overnight culture, which had been grown individually in LB medium at 37 °C with shaking at 170 rpm. Single-use aliquots were mixed in a 1:1 ratio with 50% glycerol, vortexed, and stored at −80 °C.

A natural wastewater community was also used to validate the main findings through an independent assay. This community was collected by centrifuging 30 L of municipal wastewater collected over 24 h (April 21, 2023) from the Ryaverket WWTP in Gothenburg, Sweden (9000 rpm, 20 min, 4 °C) using a Beckman J2-21M/E high-speed centrifuge (Beckman, Brea, USA). The resulting pellet was resuspended in 25% glycerol and stored in aliquots at −80 °C.

The synthetic *E. coli* community was exposed to each sterile-filtered wastewater sample, and selective effects were detected by plating on agar plates supplemented with different antibiotics, as described below. At the start of each selection experiment, 15 mL Falcon tubes were filled with 2.7 mL sterile-filtered wastewater and 0.3 mL LB medium. Physiological saline (0.85% NaCl) was used as the negative control, while both the "Hospital WW Sweden" and a ciprofloxacin solution were served as positive controls. The ciprofloxacin solution was used at a final concentration of 10 μg/L in 10% LB medium prepared with physiological saline. Each test tube was inoculated with 40 μL of *E. coli* mix or 62 μL of the natural wastewater community to achieve a final *E. coli* population density of ca $5 \times 10^5$ CFU/mL. The microbial population levels of the community stock solutions were determined through quantitative plating (37 °C for 20–24 h) prior to inoculation using cation-adjusted Mueller-Hinton agar plates. After vortexing, a small sample (100 μL) from the saline control tube was promptly transferred onto agar plates with or without antibiotics to assess the %resistance at the start of the selection assay (0-hour, baseline). The tubes were incubated at 37 °C with shaking at 170 rpm on a MaxQ 6000 shaker (Thermo Scientific, San Jose, USA). After 24 h, 3 μL of the grown cultures were transferred to fresh tubes containing 3 mL of the same wastewater-to-LB ratio sample as previously described. Following three passages in liquid medium (72 h in total), the outgrown cultures were plated onto agar with and without antibiotics and incubated at 37 °C for 20–24 h. Subsequently, colony counts were enumerated on cation-adjusted Mueller-Hinton plates in triplicate. Four major classes of antibiotics were tested, considering their widespread use and the occurrence of corresponding resistance in the environment: aminoglycosides (tobramycin), beta-lactams (amoxicillin/clavulanic acid, cefotaxime), fluoroquinolones (ciprofloxacin), and folate pathway antagonists (sulfamethoxazole/trimethoprim). The antibiotic concentrations on the agar plates matched the EUCAST clinical breakpoints outlined in Supplementary Data 9, with the exception of ciprofloxacin (1 mg/L) and tobramycin (4 mg/L). These two slightly modified concentrations were used based on a previously used protocol for environmental resistance selection studies[26]. A

simplified diagram illustrating the overall experimental design is presented in Supplementary Fig. 18. Due to the substantial number of samples, the selection experiments were organized into four sequential runs over a 21-week period. Each run was divided into four discrete experiments, with each experiment processing 9–15 samples in parallel. Every experiment included standardized controls: ciprofloxacin (10 μg/L) as a positive control, and saline as a negative control. Accordingly, all 49 municipal wastewater samples and one hospital wastewater sample ("Hospital WW Sweden", extra positive control) were tested within a single complete run. The entire experimental sequence—comprising all four runs—was independently repeated four times to ensure reproducibility.

The selection experiments with the natural wastewater community were performed similarly as above, but with CHROMagar™ ECC instead of cation-adjusted Mueller-Hinton, both for assessing inoculum size and for readout of the proportion of resistant *E. coli*. Six wastewater samples (chosen based on the resistance selection results from the synthetic *E. coli* community and antibiotic concentration data) were investigated. These samples (Benin, Lebanon, Madagascar, Nigeria, South Africa, and the United Arab Emirates) together with ciprofloxacin (10 μg/L), "Hospital WW Sweden", and saline controls as described above were examined in a single selection experiment. Four independent, repeated experiments (on different days) were performed.

## Quantification of selection potential

To quantify the wastewater samples' selection potential, we calculated the median colony counts on antibiotic-containing and antibiotic-free Mueller-Hinton or CHROMagar™ ECC plates (depending on the community used) from the triplicate plating for each experiment. The quotient of these counts, $\frac{CFU/mL\ on\ antibiotic-containing\ plates}{CFU/mL\ on\ control\ plates\ without\ antibiotics} \times 100\%$, termed %resistance, served as our comparative measure across samples. A higher %resistance value (wastewater/positive control, 72-h) compared to the control experiment (saline, 0-h) indicates a positive selection potential, while a lower %resistance value indicates a selection against resistance or deselection of resistance.

To assess which samples significantly exhibited selective or deselective effects, both Poisson and negative binomial regression models with log link functions were evaluated. These generalized linear models are designed for count data analysis, with the key distinction that negative binomial regression accommodates overdispersion, while Poisson regression assumes a strict one-to-one mean-variance relationship. Both models can analyze rates like %resistance by incorporating an offset term. For each antibiotic, we fitted a model according to Eq. (1).

$$log(E[count_i]) = \beta_0 + \beta_1 X_{i1} + \beta_2 X_{i2} + \cdots + \beta_k X_{ik} + log(MHcount_i) \quad (1)$$

Here, $count_i$ represents the median colony count on antibiotic-containing plates for each experiment; $MHcount_i$ represents the median colony count on antibiotic-free plates for each experiment (included as an offset); $\beta_0$ is the intercept, representing the log of % resistance in the reference experiment; $\beta_1...\beta_k$ are regression coefficients for each sample $k$, and represent the log difference in %resistance between any sample and the reference; $X_{i1}...X_{ik}$ are dummy variables indicating which sample an experiment is associated with. Rearranging Eq. (1) by moving $log(MHcount_i)$ to the left side and exponentiating both sides, Eq. (2) is obtained, where the left side is now recognized as representing %resistance. Equation (2) shows that $e^{\beta_0}$ corresponds to the %resistance in the reference experiment, $e^{\beta_0 + \beta_k}$ corresponds to the estimated %resistance for a sample $k$ and each $e^{\beta_k}$ corresponds to the ratio of %resistance in the sample $k$ *versus* the reference—the selection potential. Estimates and confidence intervals for these transformations of regression coefficients were obtained

from the fitted models.

$$\frac{E[count_i]}{(MHcount_i)} = e^{\beta_0} \times e^{\beta_1 X_{i1} + \beta_2 X_{i2} + \cdots + \beta_k X_{ik}} \qquad (2)$$

To determine which samples had significantly different %resistance from the reference for evidencing a significant selection or deselection of resistance, we tested which differences $\beta_k - \beta_0$ significantly differed from zero. Two-sided Wald tests were performed to generate $p$ values for these comparisons. To account for multiple comparisons, we applied the Benjamini–Hochberg adjustment to control the false discovery rate (FDR) for each tested antibiotic individually. Adjusted $p < 0.05$ were considered significant.

The regression models provided estimates and confidence intervals of %resistance for each sample across all five tested antibiotics, as well as a statistical assessment of each sample's selection potential. For the synthetic *E. coli* community assay, we analyzed data from 49 samples representing 47 countries, two positive controls [ciprofloxacin (10 μg/L) and "Hospital WW Sweden"], and %resistance in physiological saline at 0-hour (prior to 3 passages) as the reference. Results from 72-hour culturing in physiological saline were also included in the models as an additional comparison. For the natural wastewater community assay, we analyzed a subset of samples from six countries along with the same control settings as the synthetic *E. coli* community assay. For compatibility, the reference (saline, 0-h) was used as the baseline in both synthetic and natural wastewater community assays. Model comparison via negative log-likelihood and AIC values clearly demonstrated that the more flexible negative binomial models provided a better fit to the data than the Poisson models for all tested antibiotics in the datasets. Therefore, all analyses were based on results from the negative binomial models.

An alternative reference (saline, 72-h) was also used for the synthetic *E. coli* community assay. In this case, a higher %resistance value (wastewater/positive control, 72-h) compared to the parallel control (saline, 72-h) indicates an "increased persistence"[30], while a lower % resistance value indicates a decreased persistence[30].

All statistical analyses were performed in R (v. 4.4.0). The negative binomial models were fitted using glm.nb from the MASS package (v. 7.3–60.2)[61], Poisson models were fitted using glm from the stats package (v. 4.4.0), and multiple comparison adjustments were made using glht from the multcomp package (v.1.4–25)[62]. Comprehensive data from the statistical modeling are available in Supplementary Data 1 and Supplementary Data 2 for the synthetic *E. coli* community assay (saline, 0-hour and 72-h reference, respectively), and Supplementary Data 3 for the natural wastewater community assay (saline, 0-hour reference). The R code for this statistical analysis is available on GitHub: https://github.com/watNoel/Global-Sewage-Project.

**Assessment of sample processing and storage effects on selection potential**
Given that the wastewater samples in most cases were collected in April or May 2021, subjected to long transportation times with occasional thawing (Supplementary Data 8), and all were stored frozen for extended periods (see the Timeline in Supplementary Fig. 18), there is concern about the potential degradation of certain selective agents, which could lead to a reduced selection potential. To assess the impact of sample processing and storage on selection potential, fresh wastewater samples were collected from the Sahlgrenska University Hospital on October 26, 2023, and immediately assessed using the synthetic *E. coli* mix as described above. Parallel sample aliquots were prepared and stored according to the Global Sewage Surveillance Project protocol[27], tested for selective potency after different processing steps (Supplementary Fig. 3), and subsequently processed for the antibiotic concentration test. Three replicates were used for this assessment.

The selection experiment described here was based on a 24-hour exposure without successive passages, whereas the globally sourced municipal wastewater samples were tested with a three-passage (72-h) assay. This methodological distinction was intentional, as we had previously identified relatively high selection potentials in this hospital wastewater sample[26], and anticipated that it would maintain significant positive selection levels, rendering extended passages unnecessary for detecting observable selection effects. As a positive control, we also included the "Hospital WW Sweden" sample from the Sahlgrenska University Hospital collected on April 28, 2022—the same control utilized in our main selection experiment.

To compare selection potential between experimental conditions and the positive control *versus* controls (saline 0-hour or 24-hour), R (v. 4.4.3) with packages rstatix (v. 0.7.2)[63], multcomp (v. 1.4-20)[62], and emmeans (v. 1.8.0)[64] were used. For each type of resistance, one-way analysis of variance (ANOVA) was performed, followed by a Tukey's Honest Significant Difference (HSD) test with an alpha level of 0.05.

**Antibiotic and antibacterial biocide analysis**
The antibiotic and antibacterial biocide concentrations were quantified using OSPE-LC-MS/MS with the general information provided in Supplementary Data 10.

For the antibiotic analysis, wastewater samples were processed following previously validated protocols[15,26]. Prior to instrumental analysis, 5 ng of isotopically labeled internal standards were added to 10 mL of sample aliquots. A detailed description of automated OSPE-LC-MS/MS analysis of antibiotics has been reported by Khan et al.[65], with specific information for the antibiotics included in this work provided in Supplementary Data 11–13.

For the antibacterial biocide analysis, a modification of the approach reported by Östman et al.[42] was employed, which includes biocides prioritized by Tysklind et al. (Tysklind et al., submitted). This analytical protocol targets multiple biocide classes, including acids, aldehydes, biguanides, phenolic compounds, polyamines, and QACs. Method validation was performed using spiked samples to determine recovery rates and LOQ for each compound (Supplementary Data 14). All analyte standards included in this work were of analytical grade (≥ 98% purity), and internal standard calibration was used for quantification.

**Regression analysis**
Linear regression analysis was conducted to evaluate the pairwise relationship between variables, including observed selection rates (% resistance) after exposure, antibiotic and biocide concentrations, ARG and BRG relative abundances identified in the ResFinder[66] and MEGARes[67] databases, respectively, and national municipal water consumption levels. Gene abundance data were derived from metagenomic sequencing analysis conducted in our recent study[29], which examined the identical wastewater samples used in the present study prior to sterile filtration. A brief description of the resistance gene quantification is presented in Supplementary Text 1. The derivations of gene abundance data for this study were performed using Python (v. 3.12) along with the Pandas (v. 2.3.2) package[68]. Municipal water consumption data for each country from 2021 (the sampling year) were sourced from The World Factbook (https://www.cia.gov/the-world-factbook/countries/), Supplementary Data 15.

For analytical purposes, antibiotics and ARGs, as well as biocides and BRGs, were categorized into corresponding classes (Supplementary Data 16). Class-level representation was achieved by summing individual chemical concentrations or gene relative abundances within each class. For chemical concentrations below the LOQ and absent gene relative abundances, we substituted the corresponding LOQ values and $10^{-5}$, respectively. All data underwent log10-transformation prior to analysis to improve normality. We acknowledge that data points with very low values, such as chemical concentrations below

LOQ, could influence interpretation and potentially result in spurious associations. These observations were included for data transparency and to present the complete distribution, though the relationships they suggested warranted additional validations. Importantly, regression-based analyses were only applied from an exploratory standpoint.

Statistical analyses and visualization were performed using R (v. 4.4.3). Linear regression models were fitted for each analyzed pair to quantify the strength and significance of the relationships. For each pairwise comparison, a scatterplot with fitted linear regression lines and 95% confidence intervals, the coefficient of determination ($R^2$ value), and $q$ value to assess the statistical significance of the relationship, is presented. Herein, the $q$ values correspond to $p$ values having been adjusted with the Benjamini-Hochberg procedure, and statistical significance was set at $q$ 0.05.

Supplementary Text 1 and Supplementary Figs. 1–18 are provided in the "Supplementary Information" file. Supplementary Data 1–16 are included in the "Supplementary Dataset" file.

### Reporting summary
Further information on research design is available in the Nature Portfolio Reporting Summary linked to this article.

## Data availability
The source data underlying the quantification of wastewater selection potential are present in Supplementary Data 17 and Supplementary Data 18 in the "Supplementary Dataset" file. The previously generated sequencing data used in this article are publicly available in the European Nucleotide Archive with project accession number PRJEB84064. Further processed metagenomic data as source data of variables in the linear regression analysis are deposited in Zenodo (https://doi.org/10.5281/zenodo.14652833)[69].

## Code availability
The code used for the core statistical analysis of selection potential, including transformations of raw data, Poisson and negative binomial regression models, has been deposited on GitHub: https://github.com/watNoel/Global-Sewage-Project. This repository contains documentation for reproducing the selection potential analysis presented in the article. In addition, it contains code and documentation for obtaining the metagenomic data used in the article.

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

## Acknowledgements

We would like to thank the collaborators from the Global Sewage Surveillance Consortium for collecting and shipping wastewater samples to the National Food Institute, Technical University of Denmark. We thank Mariska Coertze (University of Gothenburg, Gothenburg, Sweden) for her help during the sterile filtration of the wastewater samples. We thank Jeremiah Nilsson (Västfastigheter Drift & Service Göteborg, Gothenburg, Sweden) for collecting effluent samples from the Sahlgrenska University Hospital (Gothenburg, Sweden) and assistance from the Ryaverket WWTP (Gothenburg, Sweden) for wastewater sampling. D.G.J.L. acknowledges the Swedish research councils FORMAS (grant No. 2021-00949) and VR (grant No. 2022-00945, 2018-05771) to D.G.J.L. D.G.J.L. and M.T. also would like to thank the EU, including the Swedish research councils FORMAS (grant No. 2020-03200) and VR (grant No. ERA-NET 243/2021) for funding, in the framework of the collaborative international consortium (BIOCIDE) financed under the ERA-NET Aquatic Pollutants Joint Transnational Call (grant No. 869178). This ERA-NET is an integral part of the activities developed by the three Joint Programming Initiatives (JPIs) on Water, Oceans, and Antimicrobial Resistance (AMR). C-F.F. acknowledges the Swedish research council FORMAS (grant No. 2021-00922). R.G. acknowledges the Czech Science Foundation (grant No. 20-04676X) and the Ministry of Education, Youth and Sports of the Czech Republic via CENAKVA project (grant No. LM2018099) for funding in the framework of the collaborative international consortium BIOCIDE.

## Author contributions

D.G.J.L. conceptualized the study, supervised it, and obtained the main funding. F.M.A. was responsible for coordinating sample collection, transfer, and logistical operations. D.A.G., M.D.E., and C-F.F. contributed methodological refinements to the experimental design and protocols. J.F., R.G., R.L., and M.T. conducted antibiotic and biocide analyses. H.M.M. compiled metagenomic sequencing data. Z.Y. and N.W. executed data analysis, statistical evaluation, and visualization of results. Z.Y. performed all selection experiments and prepared the initial manuscript draft with critical input from D.G.J.L. All authors participated in the review process and contributed to the final manuscript.

## Funding

## Competing interests

The authors declare no competing interests.
