## [Transparent Peer Review file · Nature Communications]

Antibiotic resistance selection and deselection in municipal wastewater from 47 countries

Corresponding Author: Professor D.G. Joakim Larsson

Editorial Note: Parts of this peer review file have been redacted as indicated to avoid any copy right infringement.

Version 0:

Reviewer comments:

Reviewer #1

(Remarks to the Author)

These are reviews for the manuscript entitled "Antibiotic resistance selection and deselection in municipal sewage from 47 countries" (#NCOMMS-25-33879) by Yu and colleagues.

L25 Is this in comparison to unfiltered samples? Please clarify. Clarity would also benefit from setting up the experimental design more before jumping into results.

L26 "Results were also validated..." Be more specific. What results? What do you mean by validated?

L29 Completely unclear sentence. Please revise.

L102 It would helpful to very briefly describe here the composition of the 340 isolate synthetic community. What traits were chosen in picking these 340 isolates and why? I also assume synthetic community means you mixed them all together...? It becomes clear later, but could be clear on the outset.

Although it took a minute and some diagramming to figure it out, in general I appreciate the experimental design of the paper. Although, considering the physiological diversity within *E. coli*, I do wonder if after mixing all the strains some strains just grow better under the conditions regardless of antibiotics, biocidals, or antibiotic resistance profiles. If so, would this bias the results in any way?

I see the flowcharts, but suggest including one in the main manuscript with way more detail that helps visualize the quantitative analysis, not just the physical workflow.

L535 Why was the median taken when it seems these triplicates provide good statistical value in determining if plate counts were significantly different between antibiotic and no-antibiotic plates?

L538 Was this multiplied times 100 to get %resistance or kept as a proportion. Either way this should be clarified.

L590 This 72-hr control seems most proper for comparison. Can you clarify why it was not used? L589 is cryptic and does little to explain. It appears that incubation with saline caused some deselection on its own, which might partially explain why deselection was so prevalent.

The discussion is quite long already, but authors should consider discussing the physiological effects of having antibiotic resistance. For example, a common genetic determinant for ciprofloxacin resistance in *E. coli* are mutations in genes essential for DNA replication. Could it be that the cost of cipro resistance is slower replication and growth, which may affect assay results? It would be interesting to know if the downsides of resistance to each of the antimicrobials tested have an effect here. This was touched on (e.g., L288), but not discussed sufficiently.

(Remarks on code availability)

Code could be better organized, made more readable, but it's great that it's there.

Reviewer #3

(Remarks to the Author)

This is a well-written paper that tackles a critical issue related to the emergence of resistance in wastewater and whether or not wastewater can drive the selection of resistant clones. It also takes an international approach, examining sewage from 47 countries. The authors are careful not to oversell their findings and clearly articulate the limitations of studies to date and the difficulties currently faced by investigators.

I found very few issues that concerned me.

Line 207 - E. coli

- One idea that occurred to me is that the samples could be examined by OSPE-LC-MS/MS at multiple timepoints to assess changes in concentration of the antimicrobials over time.

- Did the samples come from solely domestic sources, or was the wastewater from CSOs?

- I also think that while I understand the need to filter the samples, this would have removed viable microbes from the milieu, which, if present, could potentially alter the results here? I don't know if this is a valid concern, but I would like to hear what the authors think. What about competition and interactions with endogenous microbes in the samples/environment? Was there any consideration of genome sequencing to identify if other non-resistance pathways were selected for? Or if there were any non-culturable populations selected for.

- Filtering the sample would also remove any solid or particulate matter. This could remove an additional environment that could select for potential resistance? What if the bacteria increasingly form a biofilm phenotype

I think there would be value in adding additional species other than E. coli.

I think there would be value in considering other assays to determine resistance

I would recommend deep-sequencing the population of E. coli - plating in itself is selective.

(Remarks on code availability)

I cannot make a recommendation as regards the code.

Version 1:

Reviewer comments:

Reviewer #1

(Remarks to the Author)

(Remarks on code availability)

Reviewer #3

(Remarks to the Author)

The authors have comprehensively and thoughtfully responded to all of our comments. I also appreciate their mentioning of additional papers and grants in process. It is clear that while not everything can be done for this manuscript, they are already working on the follow-up and additional directions. I look forward to reading their future papers!

(Remarks on code availability)

Point-by-point response to the reviewers' comments

Manuscript: NCOMMS-25-33879

Title: Antibiotic resistance selection and deselection in municipal wastewater from 47 countries

Authors: Zhuofeng Yu, Declan A. Gray, Jerker Fick, Noel Waters, Richard Lindberg, Roman Grabic, Mats Tysklind, Mutshiene Deogratias Ekwanzala, Hannah-Marie Martiny, Carl-Fredrik Flach, Frank M. Aarestrup, D. G. Joakim Larsson

The **Line numbers in the following text refer to the document of “Manuscript (revised version with track changes)”.*

Response to Comments by the Editor and Reviewers

Reviewer #1 (Remarks to the Author)

Comment 1: L25 Is this in comparison to unfiltered samples? Please clarify. Clarity would also benefit from setting up the experimental design more before jumping into results.

Answer: We agree that this was unclear. This sentence specifically refers to a comparison to baseline levels (line 26, revised version with track changes), that is the ratio of resistance at the onset of the selection experiment. We have also made comparisons to parallel saline controls (% resistant bacteria at the end of exposure) with similar results. In the existing literature on resistance selection, both types of controls are found, and they are interpreted somewhat differently (please see also response to comment 9). We have therefore chosen to present a comparison to both types of controls here (please compare the new Fig. 2 and Supplementary Fig. 1). Fortunately, regardless of what control we compare with, the take-home message is the same. We have now added “compared to baseline” in the sentence in the abstract. Unfortunately, this tiny addition makes the abstract too long (it was already 199 words, and only 200 words are allowed). But we have been able to remove a couple of words in other places to make the abstract fit within the word limit.

We certainly understand the comment on the benefit of describing the experimental design in as large depth as possible in the abstract before providing the results. Given the current word-limit, we have done our best to balance aspects related to introduction, methods, results and discussion in the abstract, but with an intentional focus on results. We have attempted to introduce the most critical experimental design components in different sentences in the abstract, often in connection with mentioning the corresponding results. If the editor thinks an expanded methodology description is critical for the understanding of the abstract, and if the word limit can be extended beyond 200 words, then we would be happy to include some more methodological detail. As we acknowledge the critical aspect of providing the reader with a good understanding of the experimental design early, we have provided such an explanation already at the beginning of the Results section (given that Methods comes after Results in articles published in Nat Comm). Also, as a response to comments five and six by the same

reviewer (see below), we have now proposed a new main figure that outlines the experimental and analytical design of the study.

Comment 2: L26 “Results were also validated...” Be more specific. What results? What do you mean by validated?

Answer: We meant that similar results in terms of both selection and deselection of resistant *E. coli* was observed also using natural wastewater communities as an alternative inoculum. We have now revised the sentence (taking into consideration the strict word count limit) to: “Similar results were generated using natural wastewater communities.” (line 27, revised version with track changes).

Comment 3: L29 Completely unclear sentence. Please revise.

Answer: The sentence the reviewer refers to reads:

“None of the 22 analyzed antibiotics could be assigned as key drivers for selection in E. coli, whereas non-species-dependent predicted no-effect concentrations were often exceeded by >10-fold, particularly for folate pathway antagonists and macrolides.”

We agree that the sentence was difficult to read, and we have therefore now simplified the sentence as follows (lines 29-32, revised version with track changes):

“None of the 22 analyzed antibiotics could be assigned as key drivers for selection in E. coli, whereas e.g. folate pathway antagonists and macrolides often exceeded predicted non-selective concentrations for other bacteria by >10-fold.”

Comment 4: L102 It would helpful to very briefly describe here the composition of the 340 isolate synthetic community. What traits were chosen in picking these 340 isolates and why? I also assume synthetic community means you mixed them all together...? It becomes clear later, but could be clear on the outset.

Answer: We have now clarified already in the results section that the strains were picked from wastewater samples based on diversity of antibiotic resistance traits.

We have also clarified that they were mixed. More details are now provided in the Methods section as well. Nat Comm (and many other journals) uses a format where the Results section precedes the Methods section. We can see a strong advantage with this format as one more quickly gets to the main message of the paper, but as the reviewer recognizes, a limitation is that details regarding methodology cannot be presented until after results are presented. Throughout, we have attempted increase readability by introducing the most critical aspects of the methodology already in the Results section, but much detail still must be presented under Methods. We are happy to see that the reviewer recognizes that it all becomes clear later.

Comment 5: Although it took a minute and some diagramming to figure it out, in general I appreciate the experimental design of the paper. Although, considering the physiological diversity within *E. coli*, I do wonder if after mixing all the strains some strains just grow better under the conditions regardless of antibiotics, biocidals, or antibiotic resistance profiles. If so, would this bias the results in any way?

Answer: We are happy to read that the reviewer appreciates our experimental design. As a response to the hint given on the challenge of rapidly grasping it, and as a response to comment six below, we have added a new main figure explaining the experimental and analytical workflow (new Fig. 1). We hope this makes the experimental design easier to grasp more quickly.

The reviewer is correct in that some of the strains will probably grow somewhat better than others, and that this may have nothing to do with their antibiotic resistance profiles (when grown without antibiotics). Still, we do not think this poses a risk for introducing any important bias with regards to the overall conclusions. Here is why: The use of a parallel saline control (72-hour saline) does compensate for strain-specific differences in growth rate that could be independent of the strains' antibiotic resistance characteristics. And notably, the selection (and deselection) was observed not only in comparison to the baseline control (0-hour), but also in comparison to the parallel saline control (see e.g. Supplementary Fig. 1). We also used a positive control (ciprofloxacin) that consistently favoured resistant *E. coli* (regardless of comparison to the baseline or the parallel 72-hour saline control). Furthermore, we used a negative control (saline) where a slight decrease in resistant *E. coli* was observed over time (in line with

resistance coming with a small, relative cost in the absence of an antibiotic selection pressure). Finally, we observed very similar results with regards to selection and deselection of resistant *E. coli* across different antibiotic classes using a completely different inoculum (natural wastewater) that also contains additional species that could interact and compete for resources. We think these lines of evidence add confidence to our conclusions.

Comment 6: I see the flowcharts, but suggest including one in the main manuscript with way more detail that helps visualize the quantitative analysis, not just the physical workflow.

Answer: We appreciate this comment and have accordingly added a new main figure outlining both the main aspects of the physical workflow and the analyses (new Fig. 1).

Comment 7: L535 Why was the median taken when it seems these triplicates provide good statistical value in determining if plate counts were significantly different between antibiotic and no-antibiotic plates?

Answer: The technical triplicates of the plates used for the readout of % resistance served primarily to ensure measurement reliability rather than to potentially increase statistical power through more observations. Medians are also more robust to outliers than are arithmetic means. The statistical power achieved was based on biological replication through four independent exposure experiments per wastewater sample (while asserting good measurement reliability by technical replication within each experiment). It would certainly have been possible to use each of the plate replicates for readout (antibiotic versus non-antibiotic plates) in the statistical analyses in a hierarchical analysis (technical plate replicates nested under the biological, experimental replication). This would be a bit more complex and would not necessarily increase the power, as scattered outliers in the big dataset generated would have greater influence with such an approach, as compared to when simply using the more robust median of the plate replicates. While we acknowledge that there might be ways that marginally may increase statistical power by taking into account each technical replicate individually, we believe our conclusions using the median of technical

replicates and using four biological replicates remain statistically sound. We think the clear patterns of statistically significant selection and deselection observed across multiple antibiotic classes suggest that our approach successfully captured biologically meaningful signals. Furthermore, the consistency between our synthetic community results and validation experiments using natural wastewater communities, both employing identical analytical approaches, provides additional confidence to that conclusion.

Comment 8: L538 Was this multiplied times 100 to get %resistance or kept as a proportion. Either way this should be clarified.

Answer: We appreciate the reviewer's attention to this aspect. It should be multiplied by 100 to get % resistance. This has now been fixed (line 642, revised version with track changes).

Comment 9: L590 This 72-hr control seems most proper for comparison. Can you clarify why it was not used? L589 is cryptic and does little to explain. It appears that incubation with saline caused some deselection on its own, which might partially explain why deselection was so prevalent.

Answer: We agree that there is little explanation given for the use of two controls (0-hour and 72-hour saline) in the methods section. But we would like to draw the reviewer's attention to the rationale for using both controls is explained in some more detail in an entire paragraph in the preceding Discussion section (lines 482-488, revised version with track changes) and briefly also in the preceding Results section (lines 171-173, revised version with track changes). This includes references to literature discussing the use of different controls in selection experiments in more detail. The classical control, as applied by e.g. highly reputed evolutionary microbiologists such as Prof Dan Andersson, is indeed the % resistance at the start of the experiment (baseline). The value and interpretation of applying two different controls has also been stressed recently by e.g. Prof Will Gazes group in Exeter (<https://doi.org/10.1016/j.watres.2024.122310>) and ourselves (<https://doi.org/10.1016/j.envint.2020.106083>). It is correctly observed that incubation with saline caused some deselection of resistant strains on its own. We acknowledge this

on lines 185-186 (revised version with track changes), noting that this deselection was smaller than for many wastewaters. We do raise two explanations for the observed deselection (lines 338-340; 405-424 revised version with track changes), one being simply that resistance is costly in the absence of an antibiotic selection pressure (may apply both to the 72-hour saline control and wastewaters), the other being specific adaptations of certain environmental strains to the wastewater environment (applicable only to the wastewaters). Both of these alternative explanations have further support in the literature, which is also cited.

Comment 10: The discussion is quite long already, but authors should consider discussing the physiological effects of having antibiotic resistance. For example, a common genetic determinant for ciprofloxacin resistance in *E. coli* are mutations in genes essential for DNA replication. Could it be that the cost of cipro resistance is slower replication and growth, which may affect assay results? It would be interesting to know if the downsides of resistance to each of the antimicrobials tested have an effect here. This was touched on (e.g., L288), but not discussed sufficiently.

Answer: We certainly agree that this is a plausible explanation, and as outlined in the answer to the previous question, it has been raised and discussed in our manuscript. The reviewer correctly recognizes that we touched on the topic based on the results (lines 338-340, revised version with track changes), but we want to stress that the explanation related to the cost associated with resistance was also further discussed in the Discussion (lines 405-424, revised version with track changes). There, we also provided references to two excellent papers that discuss this phenomenon specifically and in depth, including examples with mutation-based ciprofloxacin resistance (Andersson, D. I. & Hughes, D. Antibiotic resistance and its cost: Is it possible to reverse resistance? *Nat Rev Microbiol* 8, 260–271 (2010); Melnyk, A. H., Wong, A. & Kassen, R. The fitness costs of antibiotic resistance mutations. *Evol Appl* 8, 273–283 (2015). Given that the discussion already is rather long (as acknowledged by the reviewer), we hope this is sufficient.

Reviewer #1 (Remarks on code availability)

Comment 1: Code could be better organized, made more readable, but it's great that it's there.

Answer: We have made an attempt to make the code more readable (lines 708-709 and 806-810, revised version with track changes). The complete R code for the

core statistical analysis, including data transformations and negative binomial regression models, is also moved from a supplementary file to a public GitHub repository at <https://github.com/watNoel/Global-Sewage-Project>.

This repository includes documentation for installing the R software and packages needed to run the analysis, as well as instructions to retrieve input data from the Supplementary Tables 17 and 18 (see “Supplementary Dataset”). We have also tested downloading scripts, data, and installing the required R packages (and versions specified) into a new environment, to check that the code runs, and output is generated as expected. We believe this will make the code more readable and the results more easily reproducible.

Reviewer #3 (Remarks to the Author)

Comment 1: This is a well-written paper that tackles a critical issue related to the emergence of resistance in wastewater and whether or not wastewater can drive the selection of resistant clones. It also takes an international approach, examining sewage from 47 countries. The authors are careful not to oversell their findings and clearly articulate the limitations of studies to date and the difficulties currently faced by investigators.

Answer: We appreciate the reviewer's positive words about our manuscript, including the value of the question addressed and its international scope. We particularly note the recognition by the reviewer that we are not trying to oversell our findings and that we clearly articulate limitations as recognized. In our view, the literature is unfortunately swamped by studies that oversell findings. Hence, not doing so here is highly intentional.

Comment 2: Line 207 - *E. coli*

Answer: Corrected (lines 250, revised version with track changes).

Comment 3: One idea that occurred to me is that the samples could be examined by OSPE-LC-MS/MS at multiple timepoints to assess changes in concentration of the antimicrobials over time.

Answer: We acknowledge the risk that antimicrobial concentrations may be reduced over time, in particular considering the global sampling, challenges associated with the logistics of transport (that at times involved accidental thawing, clarified for all samples in the supplementary material of the parallel paper describing the sampling by Martiny et al, in review in Nat Comm), processing both in Denmark and Sweden including thawing, freezing and transport, as well as additional storage until analyses. The consequence of such reductions in concentrations over time would be that we might underestimate the selective potency of the fresh wastewaters. As described under the heading “*Effects of sample processing and storage on selection potential*” (lines 214-239, revised version with track changes) we describe how we investigated the potential reduction by mimicking the sample pipeline (from wastewater sampling to analysis)

using a fresh sample of hospital wastewater from Gothenburg, both by studying selection potential at different steps with bacterial assays, and by measuring antibiotic concentrations by OSPE-LC-MS/MS (as suggested by the reviewer, Supplementary Fig. 3). This had to be done on a fresh sample, hence the sample from Gothenburg rather than any of the 49 main samples in the study. And, to enable detection of potential reductions in selection potential in a bacterial assay over time, it had to show some detectable level of selection when analysed fresh (hence taking a hospital wastewater sample rather than a municipal wastewater sample). We indeed observed some reduction in selection potential along the sample processing pipeline/over time (Supplementary Fig. 4). In parallel, we monitored antibiotic concentrations by OSPE-LC-MS/MS, suggesting gradual but moderate reductions in the concentrations of most antibiotics (Supplementary Fig. 5, Supplementary Table 4). Our conclusions from these analyses are (quoted from the manuscript): *Together, these suggest that while our measurements of selection potential in the sewage samples might be conservative estimates due to potential antibiotic degradation during storage, any findings of positive selection likely represent real selection potential of such wastewaters. Accordingly, it is possible that deselection could have been less pronounced in some cases if fresh samples had been available and tested.* We think that is a fair and balanced interpretation, and hope that the reviewer and editor agree.

Comment 4: Did the samples come from solely domestic sources, or was the wastewater from CSOs?

Answer: This is a highly relevant remark. The Global Sewage Surveillance Project protocol specified that sample collection should be conducted at inlets of municipal wastewater treatment facilities receiving predominantly domestic sewage sources. However, for some samples, there is limited metadata that can confirm this; for some, it is noted that the sample wastewater also receives contributions from, e.g., industries, hospitals, and slaughterhouses (Supplementary Table 8). Whether they are combined sewers (CS) that also receives storm water (rainwater from e.g. streets) is not stated, but it is expected as this is the practice in many countries (including Europe, and it is certainly the case at the investigated Swedish site). We have now explicitly pointed to the supplementary file in Martiny's parallel study describing the metadata for each sample (lines 539-541, revised version with track changes). We have also included a statement that the exact sources of selective agents in those samples that show selection

cannot with certainty be allocated to emissions from households, as the investigated municipal wastewaters in most if not all cases also receive contributions from other sources, such as hospitals, industries, slaughterhouses and stormwater (lines 517-519, revised version with track changes). We have also, throughout the entire paper including the title, exchanged the term municipal “sewage” to municipal “wastewater” as we feel it is a bit more inclusive and less claiming with regards to source.

Comment 5: I also think that while I understand the need to filter the samples, this would have removed viable microbes from the milieu, which, if present, could potentially alter the results here? I don't know if this is a valid concern, but I would like to hear what the authors think. What about competition and interactions with endogenous microbes in the samples/environment? Was there any consideration of genome sequencing to identify if other non-resistance pathways were selected for? Or if there were any non-culturable populations selected for.

Answer: In order to test the ability of the “chemical soup” present in the wastewaters to select for resistant bacteria (our aim), we needed to filter out all pre-existing bacteria as they would interfere with the assay, as correctly recognized by the reviewer. Notably, the resistance characteristics of the endogenous sewage community have been sequenced and presented in another study (Martiny et al, in review in Nat Comm) and are here used for a correlation analysis/comparison to observed selection potential. Note that high abundance of resistance genes in the pre-existing bacterial community in a given sample could either be a consequence of resistance selection within the wastewater (before sampling) of strains carrying such genes, and/or (most likely) a consequence of high level of resistance already in the microbiota of the local human population that contributes with sewage to that particular sample (see e.g. Karkman A, Berglund F, Flach CF, Kristiansson E, Larsson DGJ. (2020). Predicting clinical resistance prevalence using sewage metagenomic data. *Communications Biology*. 3:711. <https://doi.org/10.1038/s42003-020-01439-6>).

We interpret the first concern as if competition and interactions (overall) with other bacteria may affect selection processes. This is indeed a relevant concern, and it is also one reason why we did not rely solely on a community assay with only *E. coli*, but also used an assay (for a subset of the samples) with inoculum from a fully complex sewage

community. The results were strikingly similar, providing confidence that conclusions drawn from the synthetic *E. coli* community are valid also in a fully complex setting.

The reviewer asks if we did any genome sequencing to identify if other, non-resistance pathways were selected for. We interpret that as if we investigated whether there were other traits than antibiotic resistance that the wastewater favoured. The simple answer is no, we did not, as it would likely be a very large (costly and time-consuming) endeavour to perform large-scale genome sequencing of sufficient numbers of sewage-adapted bacteria (as well as appropriate controls) to draw firm conclusions about such an open question. Thus, we find it well beyond the scope to do it within this study. Still, we certainly think the question is valid, and it does relate to our speculation that there could be other wastewater-adaptations that are important for fitness in wastewater environments (as discussed in e.g. lines 426-433, revised version with track changes). We might add that the senior author of this study (Larsson) currently has a research grant proposal under evaluation that more openly explores if there are any wastewater adaptations in sewage *E. coli* (in addition to antibiotic resistance) using Genome Wide Association Studies (GWAS) for +1000 isolates, as well as other complementary approaches, and if so, identify which these adaptations are. So, given the application is successful, maybe we will be able to come back to this question in three or four years from now!

The last question raised was if there were any non-culturable populations selected for. We interpret this question a bit wider, basically, if we can conclude whether there is a resistance selection potential in other species than *E. coli*, whether they are cultivable or not (see also comment 7 below). This is certainly also a highly valid question. We have pointed out in several places that we only investigate selection potential in *E. coli*, and we cannot exclude that in samples that show no selection potential in *E. coli*, they might be selective in other species, at least those with different background sensitivity to various antibiotics (lines 466-468 and 478-480, revised version with track changes). One might think that metagenomic analyses and counting of ARGs in exposed/control communities would be an ideal way to extend the scope to investigate selection beyond *E. coli* (as it would still involve an exposure/cultivation step in the presence of wastewater, it would still exclude those species that cannot be cultivated in available media). However, such an approach does not provide conclusive evidence about

resistance selection (i.e. the favouring of resistant strains within a species over non-resistant counterparts). Basically, any change in species composition as a result of the exposure (also unrelated to acquired antibiotic resistance) will have consequences for ARG counts, because different species tend to carry different ARGs to different extents. So, while metagenomic sequencing can provide some kind of “overall” picture that includes all species in the sample, it also suffers from the critical drawback that one cannot conclusively draw conclusions about resistance selection in ANY given species in the community! That is why we here choose to focus on a cultivation-based assay, despite its apparent limitations. Please also see the reply to comment 7.

Comment 6: Filtering the sample would also remove any solid or particulate matter. This could remove an additional environment that could select for potential resistance? What if the bacteria increasingly form a biofilm phenotype.

Answer: Sterile-filtering was considered a good approach to remove existing bacteria, which was necessary for interpretation of results, as also recognized by the reviewer (previous comment). We have successfully used the filtering approach in the past (*Kraupner N, Hutinel M, Schumacher K, Gray DA, Genheden M, Fick J, Flach CF, Larsson DGJ. (2021). Evidence for selection of multi-resistant E. coli by hospital effluent. Environment International. 150:106436. <https://doi.org/10.1016/j.envint.2021.106436>*), but an alternative approach, at least in theory, could have been to sterilize samples with e.g. UV (which on the other hand can biotransform many chemicals into more toxic counterparts). We acknowledge that the removal of particles could influence selection, in particular we have been considering the risk that antimicrobials bind strongly to particles and are removed in the filtering step (discussed for QACs on lines 388-393, revised version with track changes). On the other hand, if selective agents are so strongly bound to particulate matter, they would likely also be less available to bacteria and hence of less importance from the perspective of risks for resistance selection. We recognize that the concern raised by the reviewer relating to the removal of particulate matter is somewhat different, basically that it would change the milieu for the bacteria in the assay from primarily planktonic growth to additional biofilm growth on particles. This is of course possible, but most literature suggest that it takes more, not less, antibiotics to select for bacteria in a biofilm compared to planktonic bacteria. We should also point out that in the complementary

assay where we use fully complex sewage as inoculum, we actually include particulate matter! This is because we generate the inoculum simply by centrifuging down bacteria (and particles) in a composite, raw wastewater sample. And we observed very similar results in that assay, as compared to with the synthetic *E. coli* community (without particles). Hence, we think the effect of not including particulate matter is probably very minor.

Comment 7: I think there would be value in adding additional species other than *E. coli*.

Answer: We agree that it would certainly be interesting to study selection in other species as well, but at present we can unfortunately not see how it could be realistically done. With regards to applying a metagenomic approach, not based on cultivation, please see our reply to comment 5 above. If one wants to use a fully complex community for a cultivation-based selection assay, it requires either that one can cultivate the species of interest/separate it from all the other species, for example by using selective agar. Here we use CHROMagar™ ECC medium for *E. coli* that has a ca 99% specificity for *E. coli* in sewage (Flach CF, Genheden M, Fick J, Larsson DGJ. (2018). A comprehensive screening of *E. coli* isolates from Scandinavia's largest sewage treatment plant indicates no selection for antibiotic resistance. *Environmental Science & Technology*. 52:11419. <https://doi.org/10.1021/acs.est.8b03354>). But, unfortunately, there is no protocol published that can identify any other bacterial species within a complex sewage community with anything near that precision (the closest one to become useful is *Klebsiella pneumoniae*; Camacho JB, Nilsson J, Larsson DGJ, Flach CF. (2024). Evaluation of culture conditions for sewage-based surveillance of antibiotic resistance in *Klebsiella pneumoniae*. *Journal of Global Antimicrobial Resistance*. 37:122-128. <https://doi.org/10.1016/j.jgar.2024.03.005>). Until such protocols are developed, there is no viable possibility to use fully complex sewage communities in selection assays for other bacterial species than *E. coli*. It would of course be possible to generate synthetic single-species communities for other species (as done here for *E. coli* using 340 strains), and it is something we are considering for the future. But it would be a huge effort to just add one more cultivable species to a study of this size (and it would still not cover uncultivable ones). And critically, we would not have sufficient volumes of wastewater samples left to test even one more species.

We do recognize the value of including additional species than *E. coli* in the future in the following, now expanded, section in the Discussion (lines 464-470, revised version with track changes):

*“However, there are certainly other species, particularly those thriving in sewage sludge and biofilms, that could be exposed to wastewater for many more generations than *E. coli*, before being released into waterways (Kristensen et al, 2021). In addition, it is also possible that resistant strains could be selected for in species that are more sensitive than *E. coli* to some of the antibiotics present in wastewater. Therefore, more studies on different bacterial species, as well as field studies reflecting the proportion of resistant bacteria in influents and effluents, would be valuable.”*

We also come back to this aspect in the second to last paragraph in the Discussion (lines 513-515, revised version with track changes):

*“Second, while we have demonstrated selection in *E. coli* as both a highly relevant pathogen and an environmental bacterium, selection dynamics are likely to differ for other bacterial species.”*

We hope the reviewer and the editor share our view that this is a fair and balanced presentation.

Comment 8: I think there would be value in considering other assays to determine resistance

Answer: We agree that the level of evidence would be/is strengthened by the use of different assays to assess selection. Indeed, this is a major reason why we applied a complementary setup to some samples. Our main setup to assess resistance selection was based on an assay using a diverse, single-species planktonic community, while the complementary setup was based on a fully complex sewage community. Both gave very similar results (previous Fig. 2, now Fig. 3). We have elaborated above (reply to comment 5) on why a metagenomic approach would not be conclusive with regard to resistance selection. We have also clarified in the manuscript that the assay applied only assesses selection in *E. coli*.

We have indeed spent much time and resources over more than a decade to consider the

pros and cons of different assays to study the selection of resistance. We were probably the very first researchers in the world to apply shotgun metagenomics to study resistance selection in the field, although at that time our understanding of what was possible to conclude and what was not was considerably less developed (*Kristiansson E, Fick J, Janzon A, Grabic R, Rutgersson C, Weijdegård B, Söderström H, Larsson DGJ. (2011). Pyrosequencing of antibiotic-contaminated river sediments reveals high levels of resistance and gene transfer elements. PLoS ONE 6:e17038. <https://doi.org/10.1371/journal.pone.0017038>. Highlighted in Nature (2011). <https://doi.org/10.1038/news.2011.46>). Last year, the senior author (Larsson) was commissioned by the World Health Organization to draft an “evidence-synthesis” on strategies/assays to derive what concentrations of different antibiotics that select and do not select for resistance (*World Health Organization. (2024). Evidence synthesis for deriving PNECs for resistance selection. Background document to “Guidance on wastewater and solid waste management for manufacturing of antibiotics”. 21 pp. Geneva. <https://www.who.int/publications/i/item/9789240097254>). Most (but not all) of the assays scrutinized (in detail) in the WHO evidence synthesis would also be applicable to determining if a complex mixture (such as wastewater) has the potential to select for resistance (the aim of the present study), with similar pros and cons as presented for antibiotics in the evidence synthesis.**

[redacted]

Above is an extract of a summary table from the WHO evidence synthesis (please see

the online document for more context and a legend: *Page 6 in “Background document: Evidence synthesis for deriving PNECs for resistance selection”*, <https://www.who.int/publications/i/item/9789240097254>). The two assays applied in the present study would correspond to the 6th and 8th column/approach in the table. Note that the MIC-approach (11th column/approach) based on available EUCAST data of thousands of clinical isolates is not applicable to assess selection in a complex mixture such as wastewater. We particularly want to draw attention to the second and third last rows in the table, indicating how confidently one can conclude within-species selection with a given assay. With this in mind, we think the chosen assays are well motivated.

It is an extensive effort to run selection assays on this scale. It took 21 weeks of full-time work to just run one assay for all wastewater samples for five antibiotic resistances, with biological and technical replication as well as testing different dilutions throughout to get reliable CFU counts (not too many, not too few). Then we are still not taking into consideration the time for collecting and preparing the synthetic community or for analysing the results. Costs for agar plates for the main applied selection assay were also several tens of thousands of USD. For the assay using a fully complex sewage community as inoculum, applied to a subset of the samples, costs per sample were roughly tripled because of the need for more expensive, selective chrome agar. Furthermore, and highly critical, for most samples, we have already used up most of the available wastewater. Taken together, we therefore hope the reviewer and the editor agree that we have made a reasonable effort with regard to the choice of selection assays, and that the coherent results of the two assays are sufficient, given that we clearly articulate that we cannot exclude selection in species not studied.

Comment 9: I would recommend deep-sequencing the population of *E. coli* - plating in itself is selective.

Answer: We are not 100% sure how to interpret the comment, but we think the comment may have been intended as a suggestion to deep-sequence the communities (metagenomics) after exposure to wastewater (and compare it to baseline/parallel saline controls). We would see the value of doing this if the inoculum is fully complex sewage (as the metagenomic analyses would expand the scope beyond *E. coli*) but we also recognize the caveats of interpretation of changes in ARG abundances in a multi-species

community, as indicated in the responses above (it is not conclusive with regards to selection of resistance in any species as it is confounded by changes in taxonomy). It would also be difficult to see a major added value of such a metagenomic effort if the reviewer's intention was a recommendation to consider it in selection experiments where the inoculum used is a defined mix of only *E. coli* (as in our main assay), rather than a fully complex multispecies community; From the phenotypic results generated, we already know whether each wastewater select or deselect for resistant *E. coli*. With metagenomic data we would, however, be able to figure out which ARGs would have increased in the *E. coli* population. But given that we in the great majority of cases see strong co-selection (or co-deselection) of resistances to several antibiotics, even in the positive control where only one, single antibiotic (ciprofloxacin) is added, knowing which ARGs are increased will still not say much about the drivers of selection! Another possible value would be that a metagenomic analysis of the *E. coli* community might point to other genes/traits that are valuable for growing in the wastewaters (see also the part of our response to comment 5 that deals with this topic). However, co-selection of various genes would most likely make any conclusions about causality highly shaky with a metagenomic, deep-sequencing approach (that is why we are considering Genome Wide Association Studies in the future, as outlined in response to comment 5). And the open question asked (looking at any gene/trait rather than simply resistance to five antibiotics) would increase the need for replication dramatically to be able to manage expected false positives. We also want to mention that we agree with the reviewer that plating itself is selective. Some strains (also within *E. coli*) will have better opportunities to grow than others under the given conditions. Still, in the defined *E. coli* community used, we know that every strain can grow on the non-antibiotic plates used. Also, the use of parallel controls in time (72-hour saline) is a way to control for the differential growth of *E. coli* strains that are unrelated to antibiotic resistance characteristics.

Finally, we might place a reminder to what was mentioned in the response to the previous comment with regards to effort, costs and limited samples available. Hence, taken together, we do not really see the necessity, nor the practical possibility, to complement the study with metagenomic analyses. We hope this is acceptable for both the reviewer and the editor.

Reviewer #3 (Remarks on code availability)

Comment 1: I cannot make a recommendation as regards the code.

Answer: OK, please see response relating to code for reviewer #1.